



# GHOSH v1.0.0: a novel Gauss-Hermite High-Order Sampling Hybrid filter for computationally efficient data assimilation in geosciences

Simone Spada[1], Anna Teruzzi[1], Stefano Maset[2], Stefano Salon[1], Cosimo Solidoro[1], and Gianpiero Cossarini[1]

[1]National Institute of Oceanography and Applied Geophysics - OGS
[2]University of Trieste

**Correspondence:** Simone Spada (sspada@ogs.it)

**Abstract.** Data assimilation is used in a number of geophysical applications to optimally integrate observations and model knowledge. Providing an estimation of both state and uncertainty, ensemble algorithms are one of the most successful data assimilation approaches. Since the estimation quality depends on the ensemble, the sampling method is a crucial step in ensemble data assimilation. Among other options to improve the capability of generating an effective ensemble, a sampling method featuring a higher polynomial order of approximation represents a novelty. Indeed, the order of the most widespread ensemble algorithms is usually equal to or lower than 2. We propose a novel hybrid ensemble algorithm, the Gauss-Hermite High-Order Sampling Hybrid (GHOSH) filter, version 1.0.0, which we apply in a twin experiment (based on Lorenz96) and in a realistic geophysical application. In the most error components, the GHOSH sampling method can achieve a higher order of approximation than in other ensemble based filters. To evaluate the benefits of the higher approximation order, a set of thousands twin experiments of Lorenz96 simulations has been carried out using the GHOSH filter and a second order ensemble Kalman filter (SEIK; singular evolutive interpolated Kalman filter). The comparison between the GHOSH and the SEIK filter has been done by varying a number of data assimilation settings: ensemble size, inflation, assimilated observations, and initial conditions. The twin-experiment results show that GHOSH outperforms SEIK in most of the assimilation settings up to a 69% reduction of the root mean square error on assimilated and non-assimilated variables. A parallel implementation of the GHOSH filter has been coupled with a realistic geophysical application: a physical-biogeochemical model of the Mediterranean Sea with assimilation of surface satellite chlorophyll. The simulation results are validated using both semi-independent (satellite chlorophyll) and independent (nutrient concentrations from an in-situ climatology) observations. Results show the feasibility of GHOSH implementation in a realistic three-dimensional application. The GHOSH assimilation algorithm improves the agreement between forecasts and observations without producing unrealistic effects on the non-assimilated variables. Furthermore, the sensitivity analysis on GHOSH setup indicates that the use of a higher order of convergence substantially improves the performance of the assimilation with respect to nitrate (i.e., one of the non-assimilated variables). In view of potential implementation of the GHOSH filter in operational applications, it should be noted that GHOSH and SEIK filters have not shown significant differences in terms of time to solution, since, as in all ensemble-like Kalman filters, the model integration is by far more computationally expensive than the assimilation scheme.





## 1 Introduction

Data assimilation (DA) methodologies provide a conceptual framework to integrate the information contents embedded in observations and numerical models, and plays a pivotal role in deriving accurate estimates of the state of earth systems components and reducing the uncertainties of geophysical systems (Bannister (2017a); Houtekamer and Zhang (2016); Edwards et al. (2015); Lahoz and Schneider (2014); Dowd et al. (2014); Fennel et al. (2022); Hersbach et al. (2020)). In fact, the use of

DA has been steadily increasing in the last decade and it is now ubiquitous in earth sciences, also thanks to the unprecedented -and always increasing- availability of both data and computational capability.

The Kalman filters and the variational approaches are the two main classes of DA algorithms (Kalman (1960), Talagrand and Courtier (1987))). Both approaches are based on Gaussian approximation and can be interpreted in a Bayesian framework, and include an estimate of the most probable state and its evolution, and possibly the associated variance. Both methods are strictly

valid only for linear models, but modified implementations have been proposed for application to non-linear ones (Carrassi et al. (2018)).

Recently, thanks to the scalability of parallel implementations, the use of ensemble algorithms (e.g., EnKF Evensen (1994) and EnVar Bannister (2017b)) have been proposed to estimate uncertainty and improve assimilation skills in Kalman filters and variational methodologies. Furthermore, some of the strong points of the two approaches have been merged, leading to hybrid

filters (e.g., Hamill and Snyder (2000)).

However, the definition of the strategy for ensemble generation is not a trivial task (Moore et al. (2019)). Straightforward Montecarlo approaches are not usually a viable option in geoscience applications, because they would require a too large number of ensemble members, and consequently, computational effort. The number of ensemble members can be reduced by adopting deterministic (or semi-deterministic) sampling methods. These methods provide an ensemble mean of future state

with no error after evolving the ensemble, as long as the evolution function (i.e., the model) is a polynomial of the same order of the sampling method. Indeed, the use of deterministic or semi-deterministic ensembles, built using second-order sampling methods, are now quite common (as in SEIK, Pham (1996), or ETKF, Bishop et al. (2001)).

At the same time, most of the models used in geoscience applications are based on systems of differential equations that cannot be represented by a second order polynomial and in all of these cases the second order sampling methods provide

only a second order approximation of the model. Furthermore, the second order approximation is more effective the closer the ensemble members are to each other, thus, the larger the ensemble spread the worse will be the approximation error in the mean computation. Since a relatively large ensemble spread is often required in data assimilation applications, this approximation error may be not negligible.

A potential strategy to reduce this error is the use of a higher order of approximation, but this would require a larger ensemble

with respect to the second order case and consequently bigger computational costs, given that a higher order of approximation implies a larger number of ensemble members to represent the same error subspace. For instance, second order methods use an ensemble of $r + 1$ members to span an $r$-dimensional error subspace (e.g., Pham (1996)), while it can be proven that $2r$





ensemble members are needed to achieve order 3 in the same subspace (an example of 3rd order method with $2r + 1$ ensemble members is the unscented Kalman filter, Wan and Van Der Merwe (2000)).

The approximation order can be increased further by using properly weighted ensemble members. A weighted ensemble based on multi-dimensional Gauss-Hermite quadrature rule (Mastroianni and Milovanovic (2008)) could arbitrarily increase the order of approximation, but always at the cost of a larger ensemble size (Ito and Xiong (2000)). Weighted ensembles are also exploited in particle filter methods (Bocquet et al. (2010)), where weights are used to evolve both the ensemble members and their probability.

In the present work, we propose a novel weighted ensemble method based on a new high-order sampling, that provides ensemble mean estimates of order higher than 2 without increasing the number of ensemble members, and therefore the computational cost.

The proposed high-order filter (as its sampling method) exploits a Gauss-Hermite-like quadrature rule, along with a principal component analysis, and we named it Gauss-Hermite high-order sampling hybrid (GHOSH) filter. Members and related

weights are chosen in such a way as to guarantee accuracy of order higher than 2 for a subset of relevant modes (a relevant subspace), and no less than 2 for the remaining ones. The GHOSH filter exploits a hybrid approach that considers the model uncertainty described by a constant and time-evolving part based on the evolution of an ensemble.

The reliability of the new filter is proven first in a large set of twin experiments based on an idealized model commonly used to test DA methodologies (Lorenz96, Lorenz (1996)) and then in the much more complex and realistic marine biogeochemical

application currently used in Copernicus Marine CMEMS System (Salon et al. (2019)), featuring assimilation of satellite observations.

The high non-linearity of the biogeochemical processes makes the chosen application a challenging framework suitable for a GHOSH test implementation. Indeed, biogeochemical DA is relatively recent both in terms of methods and assimilated variables (Fennel et al. (2019)). Surface chlorophyll concentration, which is estimated by satellite ocean colour observations,

is the most commonly assimilated biogeochemical variable since its relatively high coverage and its near real-time availability make it suitable also for operational applications (e.g., Song et al. (2016); Tsiaras et al. (2017); Santana-Falcón et al. (2020); Pradhan et al. (2019); Teruzzi et al. (2018b); Ciavatta et al. (2016)).

Section 2 introduces the novel elements of the high-order sampling and Section 3 presents a synthetic description of the GHOSH algorithm and its localized version. Additional mathematical details are provided in Appendix A. The two sets of

numerical experiments (a twin experiment of Lorenz96 model and a realistic 3D marine biogeochemical application) are introduced (Section 4) and their results presented (Section 5). The algorithm and the experimental results are discussed in Section 6.

## 2 The high-order sampling

In this work we propose a novel method to sample the ensemble members in data assimilation applications. Compared to de-

terministic square-root sampling methods (Carrassi et al. (2018)), the novel strategy achieves a higher order of approximation

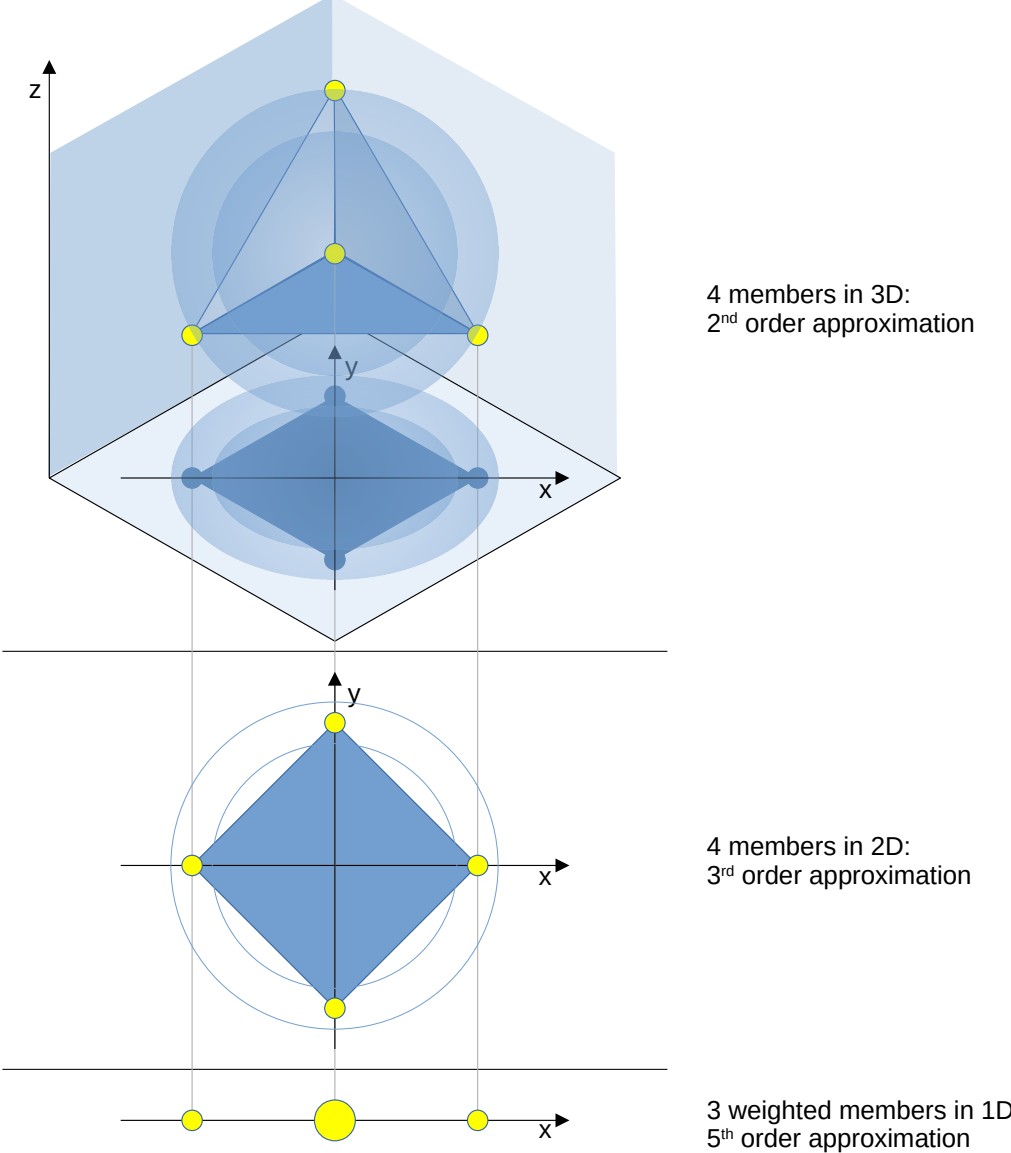

4 members in 3D:
2nd order approximation

4 members in 2D:
3rd order approximation

3 weighted members in 1D:
5th order approximation

**Figure 1.** 4 tetrahedron-shaped ensemble members (yellow dots) in a 3-dimensional space (top) represent a second-order approximation of a standard normal distribution (in shadow the spherical isosurfaces). The same 4 members form a square in 2 dimensions (middle), producing a third-order approximation. By projecting further into 1 direction (bottom), they become a fifth-order weighted ensemble.

without increasing ensemble size. The strategy is based on the choice, among all the second order ensembles, of those that feature a higher order of approximation along the most relevant components of the error (i.e., those directions where the ensemble spread is larger). To better understand the idea, Fig. 1 shows an error represented by a 3-dimensional standard normal





distribution (in shadow the spherical errors isosurface). Using the second order exact sampling (Pham (1996)), 4 ensemble
members (yellow points) are needed to represent the uncertainty produced by this error distribution. In a $xyz$ Cartesian coordinate system, the 4 ensemble members are the vertices of a regular tetrahedron, and all the possible second order samplings correspond to a rotation of this tetrahedron. With the appropriate rotation, the vertices draw (project) a square in the $xy$ plane. In this way, we achieved a third order approximation in the $xy$ plane, because it can be proved that 4 square-shaped ensemble members provide a third order approximation for a 2-dimensional normal distribution. Similarly, a further projection on a single axis returns 4 points or, as in the figure, only 3 points, one of which represents 2 of the original points. In the 1D case, the highest approximation order that can be obtained using 4 members is still limited to 3, unless proper weights are assigned to each ensemble member. In fact, 4 weighted members (or even only 3) are sufficient to reach an approximation order as high as 5 on the $x$ axis. Finally, what we have obtained is a 3-dimensional second order weighted ensemble which is a third-order approximation in the $xy$ plane and a fifth order approximation along the $x$ axis.

This means that, if the $z$ component of the distribution was less relevant than the $x$ and $y$ components (e.g., a non-standard normal distribution with smaller variance on the $z$ coordinate), then we obtained a third order approximation in the dimensions where the spread is higher. And, if the variance is bigger on $x$ than $y$, then we achieved a fifth order approximation where the spread is maximum.

The idea described above can be generalized considering that any probability distribution (under reasonable hypothesis) can be sampled with an arbitrary high order of approximation with a weighted ensemble that, in the special case of the Gaussian distribution, is represented by the nodes and weights of the Gauss-Hermit quadrature rule (Mastroianni and Milovanovic (2008)). Once the approximation order ($h$ larger than 2) and the ensemble size ($r+1$) are fixed, the $h$-order weighted ensemble is computed for a relatively small $s$-dimensional subspace ($s$ lower than $r$); then, a new $(r+1)$-sized second order weighted ensemble is computed such that its projections along the most relevant error directions coincide with the $h$-order ensemble. The resulting ensemble has a second order approximation in the $r$-dimensional subspace and a $h$-order approximation in the $s$-dimensional subspace of the most relevant error directions. By comparison, a second order exact sampling (e.g., SEIK) with $r+1$ ensemble members would provide a second order approximation for the whole $r$-dimensional subspace.

The high-order sampling is described in the following from an algorithmic point of view. Mathematical details, instead, can be found in Appendix A.

The high-order sampling presents two phases: *initialization* and *sampling*. The former includes the steps that must be executed only once, while the latter presents the ensemble generation procedure.

## 2.1 Initialization

First, 3 hyper-parameters $r$, $h$ and $s$ are chosen: $r$ is the dimension of the error subspace, which implies an ensemble size of $r+1$ members; $h$ is the higher approximation order reached in the most relevant directions; $s$ represents the number of principal components that are approximated with order $h$ and must be smaller than $r$.

Then, the real numbers $u_j^m$ and $v_j$, with $j \in \{1, \dots, s\}$ and $m \in \{1, \dots, r+1\}$, are calculated as a solution of the nonlinear





system

$$
\begin{cases}
\vdots \\
\sum_{m=1}^{r+1} u_{j_1}^m \cdots u_{j_\xi}^m v_m^{\,2-\xi} = \mu_{j_1,\ldots,j_\xi}, \\
\vdots
\end{cases}
\tag{1}
$$

with one equation for each $\xi \in \{0,\ldots,h\}$ and for each $j_1,\ldots,j_\xi \in \{1,\ldots,s\}$. $\mu_{j_1,\ldots,j_\xi}$ are the statistical moments of a $s$-
dimensional probability distribution that approximates the assumed error distribution shape. Such probability distribution must
be uncorrelated, normalized and have $0$ mean. A typical choice is the Normal distribution $\mathcal{N}(\boldsymbol{z};0,\mathbf{I}_s)$, i.e.,

$$
\mu_{j_1,\ldots,j_\xi} = \int_{\mathbb{R}^s} z_{j_1} \cdots z_{j_\xi} \mathcal{N}(\boldsymbol{z};0,\mathbf{I}_s)\,d\boldsymbol{z}.
\tag{2}
$$

In the Gaussian case, the moments $\mu_{j_1,\ldots,j_\xi}$, which are other hyper-parameters of the GHOSH algorithm, can easily be com-
puted (i.e., without solving numerically the integral, see Appendix A, equations (A6) and (A7)). Changing the value of the
moments $\mu_{j_1,\ldots,j_\xi}$ allows to explore other non-Gaussian probability distributions.

Note that, in order to actually have at least a solution of the system, the number of independent equations cannot be bigger than
the number of variables, which implies that a bigger $h$ requires a smaller $s$ or a bigger $r$.

The system solution is used to define $\boldsymbol{\Omega}_h$, as the $(r+1) \times s$ matrix with coordinates $u_j^m$, and the ensemble weight vector
$\boldsymbol{w} \in \mathbb{R}^{(r+1)}$, as the element-wise square of $\boldsymbol{v}$, i.e.,

$$
\boldsymbol{w} = \left(v_1{}^2,\ldots,v_{r+1}{}^2\right).
\tag{3}
$$

## 2.2 Sampling

Given an $N$-dimensional state space, in the sampling phase a new ensemble is generated based on the ensemble mean $\boldsymbol{x}_i \in \mathbb{R}^N$
and the covariance matrix $\mathbf{L}_i\mathbf{L}_i^T$. Differently from non-varying quantities (like those defined in the *initialization* phase), $i$-
indexed quantities are specific of each sampling. The columns of the $N \times r$ matrix $\mathbf{L}_i$ adopted to obtain a $r$-rank factorization
of the covariance matrix represent the base of the error subspace spanned by the ensemble members.

A random $s \times s$ orthogonal matrix $\boldsymbol{\Omega}_i^{rnd}$ is drawn, then the $(r+1) \times (s+1)$ matrix $\begin{pmatrix} \boldsymbol{v} & | & \boldsymbol{\Omega}^h\boldsymbol{\Omega}_i^{rnd} \end{pmatrix}$ is completed to
an $(r+1) \times (r+1)$ orthogonal matrix by the $(r+1) \times (r-s)$ matrix $\boldsymbol{\Omega}_i^\perp$, i.e.,

$$
\left(\boldsymbol{v} \;\middle|\; \boldsymbol{\Omega}^h\boldsymbol{\Omega}_i^{rnd} \;\middle|\; \boldsymbol{\Omega}_i^\perp\right)^T \left(\boldsymbol{v} \;\middle|\; \boldsymbol{\Omega}^h\boldsymbol{\Omega}_i^{rnd} \;\middle|\; \boldsymbol{\Omega}_i^\perp\right) = \mathbf{I}_{r+1},
\tag{4}
$$

where $\boldsymbol{v}$ is the array of the square roots of the weights (as in equation (3)) and $\mathbf{I}_{r+1}$ is the identity matrix of rank $r+1$. Many
procedures (e.g., Pham (1996)) are available to generate a random orthogonal matrix (or to randomly complete a given set of
orthonormal vectors to an orthogonal matrix).



Now, the $(r+1) \times r$ matrix $\mathbf{\Omega}_i$, defined as

$$\mathbf{\Omega}_i = \left( \begin{array}{c|c} \mathbf{\Omega}^h \mathbf{\Omega}_i^{rnd} & \mathbf{\Omega}_i^{\perp} \end{array} \right), \tag{5}$$

can be used to build the $N \times r$ ensemble matrix $\mathbf{X}_i$:

$$\mathbf{X}_i = \boldsymbol{x}_i \mathbb{1}_{1 \times (r+1)} + \mathbf{L}_i \mathbf{C}_i{}^T \mathbf{\Omega}_i{}^T \mathbf{W}^{-\frac{1}{2}}, \tag{6}$$

where $\mathbb{1}$ is a matrix (of the subscripted size) filled with ones, $\mathbf{W} = \operatorname{diag}(\boldsymbol{w})$ is the $(r+1) \times (r+1)$ diagonal matrix of weights and $\mathbf{C}_i$ is a $r \times r$ orthogonal change-of-basis matrix such that

$$\mathbf{L}_i{}^T \mathbf{\Lambda}_i{}^{-1} \mathbf{L}_i = \mathbf{C}_i^T \mathbf{E}_i \mathbf{C}_i. \tag{7}$$

In the last equation, the right-hand side is an eigenvalue decomposition of the left-hand side, with $\mathbf{E}_i$ being an $r \times r$ diagonal matrix of eigenvalues in descending order and $\mathbf{\Lambda}_i$ being an appropriate $N \times N$ matrix. The purpose of $\mathbf{\Lambda}_i$ is to weight the product between $\mathbf{L}_i$ and its transposed and therefore to define the criterion that decides the relevance of each PCA component. In the simplest cases, $\mathbf{\Lambda}_i$ can be the identical matrix, but in most scenarios, $\mathbf{\Lambda}_i$ needs to be properly designed (see sections 4.2.3 and 6).

After $\mathbf{X}_i$ computation, the ensemble members can be retrieved from the column of the matrix, i.e.,

$$\mathbf{X}_i = \left( \boldsymbol{x}_i^1, \ldots, \boldsymbol{x}_i^{(r+1)} \right). \tag{8}$$

## 3 The GHOSH filter

Similar to other data assimilation ensemble schemes, the GHOSH filter provides an estimate of the state of a system at some time $t_i$ in terms of the state vector and the covariance matrix that represents the error estimate of the state vector. Hence, at time $t_i$, a forecast is produced in the form of the forecast state vector $\boldsymbol{x}_i^f \in \mathbb{R}^N$ ($N$ is the dimension of the state space) and the forecast error covariance matrix $\mathbf{P}_i^f$. Moreover, if an observation vector $\boldsymbol{y}_i$ is available at time $t_i$, the information is assimilated in the analysis state vector $\boldsymbol{x}_i^a$ with error covariance matrix $\mathbf{P}_i^a$.

Being an ensemble-based filter scheme, the GHOSH filter represents these quantities by an ensemble of state vectors

$$\mathbf{X}_i^a = \left( \boldsymbol{x}_i^{a,1}, \ldots, \boldsymbol{x}_i^{a,(r+1)} \right) \tag{9}$$

of $r+1$ model state realizations. However, differently from other ensemble filters that uniformly weight the ensemble members, the GHOSH filter assigns to $\mathbf{X}_i^a$ a vector of corresponding weights

$$\boldsymbol{w} = \begin{pmatrix} w_1 \\ \vdots \\ w_{r+1} \end{pmatrix}. \tag{10}$$





The state estimate is given by the weighted mean

$$\boldsymbol{x}_i^a = \sum_{j=1}^{r+1} \boldsymbol{x}_i^{a,j} w_j = \mathbf{X}_i^a \boldsymbol{w}, \tag{11}$$

while the error covariance is approximated by the ensemble covariance matrix

$$\mathbf{P}_i^a \approx \mathbf{X}_i^a \mathbf{W} \, \mathbf{X}_i^{a\,T} - \boldsymbol{x}_i^a \, \boldsymbol{x}_i^{a\,T}, \tag{12}$$

with $\mathbf{W} = \mathrm{diag}\,(\boldsymbol{w})$ being the diagonal matrix of weights.

The GHOSH algorithm consists of 5 phases, i.e., *initialization*, *forecast*, *forecast high-order sampling*, *analysis* and *analysis high-order sampling* (Fig. 2). The *initialization* is performed once at the beginning of the filter application, the *forecast* and

*forecast high-order sampling* phases are carried out at each time $t_i$, and *analysis* and *analysis high-order sampling* are executed only when an observation is available.

The *analysis* equations are a novel weighted version of the ensemble Kalman filter equations (Evensen (1994)), while the *forecast* phase equations rely on a hybrid approach. In fact, the forecast uncertainty is obtained by combining the ensemble covariance (weighted by a forgetting factor) and a model error covariance matrix $\mathbf{Q}_i$, which is built from the existing knowledge of the system, as done for the background matrix in many variational schemes (Bannister (2017a, 2008)).

The method includes two resampling phases, after *analysis* and *forecast* (Fig. 2), at which the whole ensemble is rebuilt using the high-order sampling method (Section 2). The resampling produces a sample of state vectors that matches the moments of the error probability density function with better precision than in the other commonly used ensemble DA algorithms. In fact, projecting the ensemble in the subspace spanned by the most relevant components of the ensemble principal components analysis (PCA), the corresponding discrete probability distribution function matches the moments of the projection of the error

uncertainty up to order $h \geq 2$ (see Appendix A). When the sampling is applied after the forecast, GHOSH allows to take further advantage of the high convergence order of the method, especially when the model error is not neglected.

### 3.1   The global GHOSH filter

#### 3.1.1   Initialization

First, following the high-order sampling *initialization* phase (Section 2.1), the three GHOSH hyper-parameters $r$ (the dimension of the error subspace), $h$ (the higher approximation order reached in the most relevant directions) and $s$ (the number of principal components that are approximated with order $h$) are used to compute the sampling matrix $\boldsymbol{\Omega}^h$ and the weights vector $\boldsymbol{w}$.

A starting weighted ensemble must be provided, with the coordinates of $\boldsymbol{w}$ as weights. Such ensemble can come from any previous forecast/assimilation using the GHOSH filter, or it can be built from scratch with an ensemble generation method

(e.g., a PCA on historical data followed by the GHOSH sampling method described in Section 2.2). The ensemble members are stored in the columns of ensemble matrix $\mathbf{X}_0^f$, i.e.,

$$\mathbf{X}_0^f = \left( \boldsymbol{x}_0^{f,1}, \ldots, \boldsymbol{x}_0^{f,(r+1)} \right), \tag{13}$$





**GHOSH flow chart**

Initialization
timestep i = 0

$X_0^f$

Observation
available?

Yes

No

$X_i^a = X_i^f$

**Analysis phase**

Observation
operator

$\mathcal{H}_i$

$Y_i$

Analysis
computation

$y_i, R_i$

Observations

$x_i^a, L_i^a$

High-order
sampling

$X_i^a$

**Forecast phase**

Model
operator,
i += 1

$\widetilde{X}_i^f$

Hybridization

$Q_i$

Model
error

$x_i^f, L_i^f$

$X_i^f$

High-order
sampling

**Figure 2.** GHOSH filter flow chart. Subscripts represent the time steps of the numerical simulation.

with the subscripted index 0 representing the first time step.

The mean $\mathbf{x}_0^f$, can be computed as

$\quad \mathbf{x}_0^f = \mathbf{X}_0^f \mathbf{w}.$ (14)

### 3.1.2 Analysis phase

If observations are available at time step $i$ (Fig. 2), $\boldsymbol{y}_i \in \mathbb{R}^n$ represents the observation array and $\mathcal{H}_i$ is the observation operator, which incorporates all operations needed to obtain the measured quantities from the state vector. This operator is used on each





ensemble member $\boldsymbol{x}_i^{f,k}$ to build the $n \times (r+1)$ matrix $\mathbf{Y}_i$, i.e.,

$$\boldsymbol{y}_i^k = \mathcal{H}_i\left(\boldsymbol{x}_i^{f,k}\right) \tag{15}$$

$$\mathbf{Y}_i = \left(\boldsymbol{y}_i^1, \ldots, \boldsymbol{y}_i^{(r+1)}\right), \tag{16}$$

with $k \in \{1, \ldots, r+1\}$.

The base $\tilde{\mathbf{L}}_i^a$ of the error subspace spanned by the ensemble is computed by

$$\tilde{\mathbf{L}}_i^a = \mathbf{X}_i^f \mathbf{T}, \tag{17}$$

where $\mathbf{T}$ is an $(r+1) \times r$ full-rank matrix with zero column sums. A reasonable choice is

$$\mathbf{T} = \begin{pmatrix} \mathbf{I}_r \\ \hline 0 \quad \cdots \quad 0 \end{pmatrix} - \boldsymbol{w}\mathbb{1}_{1 \times r}, \tag{18}$$

with $\mathbf{I}_r$ being the identity matrix of size $r$ and $\mathbb{1}$ is a matrix (of the subscripted size) filled with ones. Equation (18) implies the use of the ensemble anomalies as basis members, and it is a common choice in SEIK implementations (e.g., Triantafyllou et al. (2003)), but other $\mathbf{T}$ can be explored without affecting the algorithm (e.g., see Nerger et al. (2012)).

The analysis error covariance matrix $\mathbf{P}_i^a$ is obtained by

$$\mathbf{P}_i^a = \tilde{\mathbf{L}}_i^a \mathbf{A}_i^a \tilde{\mathbf{L}}_i^{a^T}, \tag{19}$$

where

$$\mathbf{A}_i^{a-1} = \mathbf{T}^T W^{-1} \mathbf{T} + \mathbf{Z}_i^T \mathbf{R}_i^{-1} \mathbf{Z}_i, \tag{20}$$

and

$$\mathbf{Z}_i = \mathbf{Y}_i \mathbf{T}, \tag{21}$$

$W = \mathrm{diag}(\boldsymbol{w})$ being the $(r+1) \times (r+1)$ diagonal matrix of weights and $\mathbf{R}_i$ the $n \times n$ error covariance matrix representing the uncertainty in the observation $y_i$.

An opportune change in basis is used to obtain the simpler decomposition




$$\mathbf{P}_i^a = \mathbf{L}_i^a \mathbf{L}_i^{aT}, \tag{22}$$

where

$$\mathbf{L}_i^a = \tilde{\mathbf{L}}_i^a \mathbf{S}_i^a \tag{23}$$

and $\mathbf{S}_i^a$ is the symmetric square root of $\mathbf{A}_i^a$, i.e.,

$$\mathbf{S}_i^a \mathbf{S}_i^a = \mathbf{A}_i^a. \tag{24}$$

Finally, the analysis state $\boldsymbol{x}_i^a$ is estimated by

$\quad \boldsymbol{x}_i^a = \boldsymbol{x}_i^f + \mathbf{L}_i^a \mathbf{S}_i^a \mathbf{Z}_i^T \mathbf{R}_i^{-1} \left( \boldsymbol{y}_i - \mathbf{Y}_i \boldsymbol{w} \right). \tag{25}$

Equation (25) differs from the usual ensemble Kalman filter equations in the use of $\mathbf{Y}_i \boldsymbol{w}$ in the last term instead of $\mathcal{H}_i \left( \boldsymbol{x}_i^f \right)$. The two are the same if the observation operator $\mathcal{H}_i$ is linear. However, in the general case, $\mathbf{Y}_i \boldsymbol{w}$, which relies on the whole ensemble instead of the ensemble mean only, is a better estimator of the expected value of the observed quantities. Moreover, ensembles generated with GHOSH's sampling method lead to a further advantage due to the higher order of approximation.

### 3.1.3 Analysis high-order sampling

The $N \times (r+1)$ ensemble matrix $\mathbf{X}_i^a = \left( \boldsymbol{x}_i^{a,1}, \dots, \boldsymbol{x}_i^{a,(r+1)} \right)$, whose columns are the ensemble members, is obtained by

$$\mathbf{X}_i^a = \boldsymbol{x}_i^a \mathbb{1}_{1 \times (r+1)} + \mathbf{L}_i^a \mathbf{C}_i^{aT} \boldsymbol{\Omega}_i^{aT} W^{-\frac{1}{2}}. \tag{26}$$

The sampling procedure is described in Section 2.2; therefore, $\mathbf{C}_i^a$ and $\boldsymbol{\Omega}_i^a$ are computed as $\mathbf{C}_i$ and $\boldsymbol{\Omega}_i$ in equation (6).

### 3.1.4 Forecast phase

In this phase, the time step index $i$ is increased by 1 and the ensemble is evolved through the model operator $\mathcal{M}_i$ (which usually represents the numerical integration of a system of differential equations), i.e., for $k \in \{1, \dots, r+1\}$,

$$\tilde{\boldsymbol{x}}_i^{f,k} = \mathcal{M}_i \left( \boldsymbol{x}_{i-1}^{a,k} \right), \tag{27}$$

$$\tilde{\mathbf{X}}_i^f = \left( \tilde{\boldsymbol{x}}_i^{f,1}, \dots, \tilde{\boldsymbol{x}}_i^{f,(r+1)} \right). \tag{28}$$





The forecast state $\boldsymbol{x}_i^f$ is estimated as the weighted mean of the ensemble by

$$\boldsymbol{x}_i^f = \tilde{\mathbf{X}}_i^f \boldsymbol{w}. \tag{29}$$

To track the uncertainty of this estimation, the basis of the error subspace is obtained by

$$\tilde{\mathbf{L}}_i^f = \tilde{\mathbf{X}}_i^f \mathbf{T}. \tag{30}$$

The covariance matrix $\mathbf{P}_i^f$ is approximated by

$$\mathbf{P}_i^f = \tilde{\mathbf{L}}_i^f \left(\mathbf{T}^T \mathbf{W}^{-1} \mathbf{T}\right)^{-1} \tilde{\mathbf{L}}_i^{f\,T} + \mathbf{Q}_i \approx \tilde{\mathbf{L}}_i^f \mathbf{A}_i^f \tilde{\mathbf{L}}_i^{f\,T}, \tag{31}$$

where $\mathbf{Q}_i$ is the $N \times N$ covariance matrix of the model error and $\mathbf{A}_i^f$ is the $r \times r$ matrix approximating the error covariance in the reduced basis expressed by $\tilde{\mathbf{L}}_i^f$. $\mathbf{A}_i^f$ can be computed in many ways, depending on the form of $\mathbf{Q}_i$ and on the chosen hybridization strategy. Here we consider the case of $\mathbf{Q}_i$ being a full rank matrix (e.g., diagonal), thus we suggest

$$\mathbf{A}_i^f = \left(\rho \mathbf{T}^T \mathbf{W}^{-1} \mathbf{T}\right)^{-1} + \left(\tilde{\mathbf{L}}_i^{f\,T} \mathbf{Q}_i^{-1} \tilde{\mathbf{L}}_i^f\right)^{-1}, \tag{32}$$

where $\rho \leq 1$ is the forgetting factor, which is added to the equation to introduce inflation (Carrassi et al. (2018)). The last term in equation (32) is one of the novel element of the present work. It represents the projection of the model error on the error subspace defined by the ensemble. This non-orthogonal projection, induced by the scalar product defined by the matrix $\mathbf{Q}_i^{-1}$, is a form of hybridization which aims to focus on the effects of the model uncertainty in the ensemble error subspace.

Since $\mathbf{Q}_i$ can have a very large size, it should be sparse or managed in a decomposed form. $\mathbf{Q}_i$ should be built accordingly to some knowledge of the system, as done, for example, for the background error covariance matrix in variational methods (Bannister (2017a, 2008)).

A new basis for the error subspace is obtained by

$$\mathbf{L}_i^f = \tilde{\mathbf{L}}_i^f \mathbf{S}_i^f, \tag{33}$$

where $\mathbf{S}_i^f$ is the symmetric square root of $\mathbf{A}_i^f$, i.e.,

$$\mathbf{S}_i^f \mathbf{S}_i^f = \mathbf{A}_i^f, \tag{34}$$

which can be computed by an eigenvalue decomposition of $\mathbf{A}_i^f$. Finally, equation (31) can be rewritten as

$$\mathbf{P}_i^f \approx \mathbf{L}_i^f \mathbf{L}_i^{f\,T}. \tag{35}$$

Writing $\mathbf{P}_i^f$ in this form is important for the consistency of the filter and is useful for the implementation of the localization (see Section 3.2).





### 3.1.5  Forecast high-order sampling

The $N \times (r+1)$ ensemble matrix $\mathbf{X}_i^f = \left( \boldsymbol{x}_i^{f,1}, \ldots, \boldsymbol{x}_i^{f,(r+1)} \right)$, whose columns are the ensemble members, is obtained by

$$\mathbf{X}_i^f = \boldsymbol{x}_i^f \mathbb{1}_{1 \times (r+1)} + \mathbf{L}_i^f \mathbf{C}_i^{f\,T} \boldsymbol{\Omega}_i^{f\,T} W^{-\frac{1}{2}}. \tag{36}$$

The sampling procedure is similar to that in Section 3.1.3; therefore, $\mathbf{C}_i^f$ and $\boldsymbol{\Omega}_i^f$ are computed as $\mathbf{C}_i$ and $\boldsymbol{\Omega}_i$ in equation (6).

Compared to data assimilation schemes with resampling only after analysis, the GHOSH filter uses this sampling phase to produce a better representation of the uncertainty. In fact, it takes into account the model error effects in equations (31)

and (32), which are otherwise neglected. If the model error is supposed to be not significant (i.e., $\mathbf{Q}_i = 0$) then the *forecast high-order sampling* phase can be widely simplified and the ensemble members can be obtained by

$$\boldsymbol{x}_i^{f,k} = \boldsymbol{x}_i^f + \frac{1}{\sqrt{\rho}} \left( \tilde{\boldsymbol{x}}_i^{f,k} - \boldsymbol{x}_i^f \right), \tag{37}$$

for $k \in \{1, \ldots, r+1\}$.

## 3.2  The local GHOSH filter

Localization is a widely used procedure that aims to avoid spurious correlations induced by an overly small ensemble while reducing the degrees of freedom of the system (Janjic et al. (2011)).

A localized version of the GHOSH algorithm is obtained by modifying some of the equations in Sections 3.1.2 and 3.1.4. Three operators $\mathcal{L}_p$, $\mathcal{L}_p^{\mathcal{H}}$ and $\mathcal{D}$ are used to improve readability. The localization operator $\mathcal{L}_p$ takes a global (array or) matrix with $N$ rows and returns its localized counterpart with $l$ rows with respect to the domain point $p$. The simplest example of the

localization operator takes just the variable values of the cells in a small radius around $p$. $\mathcal{L}_p^{\mathcal{H}}$ works in the same way in the observation space, reducing the number of rows from $n$ to $l'$. The delocalization operator $\mathcal{D}$ takes a set of local matrices (or arrays), ideally one for each point of the domain, and returns the global counterpart. Building the global matrix by taking the values at the central points of each local matrix is a common example of the delocalization operator.

### 3.2.1  Local forecast

Instead of equations (31-34), the localized version of the forecast step of the algorithm computes, for each point $p$ of the domain, the local covariance matrix $\mathbf{A}_i^{p,f}$ by

$$\mathbf{A}_i^{p,f} = \left( \rho \mathbf{T}^T \mathbf{W}^{-1} \mathbf{T} \right)^{-1} + \left( \mathcal{L}_p \left( \tilde{\mathbf{L}}_i^f \right)^T \mathbf{Q}_i^{p-1} \mathcal{L}_p \left( \tilde{\mathbf{L}}_i^f \right) \right)^{-1}, \tag{38}$$

where $\mathbf{Q}_i^p$ is the $l \times l$ model error covariance matrix at the point $p$.

The global basis is built by changing the basis and delocalizing, i.e.,





$$\mathbf{L}_i^f = \mathcal{D}\left(\left\{\mathcal{L}_p\left(\tilde{\mathbf{L}}_i^f\right)\mathbf{S}_i^{p,f}\right\}_p\right), \tag{39}$$

where $\mathbf{S}_i^{p,f}$ is the symmetric square root of $\mathbf{A}_i^{p,f}$. Note that the change in basis induced by $\mathbf{S}_i^{p,f}$ is continuous (see, e.g., Nerger et al. (2012)); thus, the delocalization operator does not need to include averaging or other smoothing procedures to avoid discontinuities.

### 3.2.2 Local analysis

The localized version of the analysis step substitutes equations (19) and (20) by calculating, for each point $p$ of the domain, the local covariance matrix $\mathbf{A}_i^{p,a}$, i.e.,

$$\mathbf{A}_i^{p,a\,-1} = \mathbf{T}^T\mathbf{W}^{-1}\mathbf{T} + \mathcal{L}_p^{\mathcal{H}}\left(\mathbf{Z}_i\right)^T\mathbf{R}_i^{p\,-1}\mathcal{L}_p^{\mathcal{H}}\left(\mathbf{Z}_i\right), \tag{40}$$

where $\mathbf{R}_i^p$ is the $l' \times l'$ observation error covariance matrix at the point $p$. This matrix can be extracted by $\mathbf{R}_i^{-1}$, but it is convenient to increase the uncertainty at the points far from $p$ (as in Nerger and Gregg (2007)). A reasonable choice is a polynomial weight function that goes smoothly to zero out of the localization radius, e.g.,

$$f\left(d\right) = \left(\left(\frac{d}{d_l}\right)^2 - 1\right)^2, \tag{41}$$

with $d$ being the distance from $p$ and $d_l$ the localization radius.

Similarly to equation (39) for local forecast, equation (23) becomes

$$\mathbf{L}_i^a = \mathcal{D}\left(\left\{\mathcal{L}_p\left(\tilde{\mathbf{L}}_i^a\right)\mathbf{S}_i^{p,a}\right\}_p\right), \tag{42}$$

where $\mathbf{S}_i^{p,a}$ is the symmetric square root of $\mathbf{A}_i^{p,a}$.

Finally, the analysis state $\boldsymbol{x}_i^a$ in equation (25) is instead estimated by

$$\boldsymbol{x}_i^a = \boldsymbol{x}_i^f + \mathcal{D}\left(\left\{\mathcal{L}_p\left(\tilde{\mathbf{L}}_i^a\right)\mathbf{A}_i^{p,a}\mathcal{L}_p^{\mathcal{H}}\left(\mathbf{Z}_i\right)^T\mathbf{R}_i^{p\,-1}\mathcal{L}_p^{\mathcal{H}}\left(\boldsymbol{y}_i - \mathbf{Y}_i\boldsymbol{w}\right)\right\}_p\right). \tag{43}$$

## 4 Experimental setup

The GHOSH filter has been tested in two sets of numerical experiments of different complexity. First, in Section 4.1, a very large set of twin experiment based on the Lorenz96 model (Lorenz (1996)) aims to evaluate the performance of GHOSH





compared with a second order filter (SEIK). The Lorenz96 model, which has a chaotic behaviour that is comparable to the one of fluid dynamics equations at a largely lower computational cost, is a standard choice for testing new data assimilation schemes (e.g., Brajard et al. (2020); Fertig et al. (2007); Gharamti (2018); Grooms and Robinson (2021); Nerger (2022)).

The second experiment aims to assess the feasibility of the GHOSH implementation in a realistic geophysical application (Section 4.2). In particular, the GHOSH filter has been coupled with the Mediterranean biogeochemical model system currently used in Copernicus Marine CMEMS System (OGSTM-BFM: Salon et al. (2019)) to assimilate ocean color satellite data.

## 4.1 Twin Experiment

The set of twin experiments is based on the Lorenz96 model with random initial conditions. In each experiment, the *truth* is provided by integrating the model for a fixed time interval. Then, at regular time intervals, observations of a subset of the system variables have been extracted by the *truth*, adding a random error to each of them to represent the observation uncertainty. The same set of observations is then used for two assimilation experiments, one using the SEIK filter (as a second order approximation filter Nerger et al. (2005)), and one using the GHOSH filter with fifth approximation order (i.e., $h = 5$, see Section 3). Both the filters are initialized from the same initial condition, chosen randomly around the *truth* initial condition. After an initial spin up, the skill of each filter is computed by averaging the root mean square error (RMSE) between the filter forecast and the *truth* before the assimilation step. RMSE evolution in time and a synthetic RMSE value for the whole simulation time has been evaluated for each filter. Then, in order to produce reliable statistics, the same procedure has been repeated $56000$ times by changing the *truth* initial conditions, observations, filters initial conditions and filters hyper-parameters such as the ensemble size and the forgetting factor.

### 4.1.1 The Lorenz96 model

The Lorenz96 model (Lorenz (1996)) is a dynamical system commonly used as a testing framework in data assimilation. The state vector $\boldsymbol{x} = (x_1, \ldots, x_N) \in \mathbb{R}^N$ is evolved according to the system of differential equations given by, for $1 \leq j \leq N$,

$$\frac{dx_j}{dt} = (x_{j+1} - x_{j-2}) x_{j-1} - x_j + F, \tag{44}$$

with periodic conditions, i.e., $x_{-1} = x_{N-1}$, $x_0 = x_N$ and $x_{N+1} = x_1$. In our implementation, the size of the state array was $N = 62$ and the forcing term was $F = 8$, which is a common value used to generate chaotic behaviour.

The equations have been implemented in python and solved numerically using SciPy's *solve_ivp* routine.

At time $t = 0$, the state vector has been initialized by adding a small perturbation ($x_N = 0.01$) to the null solution $\boldsymbol{x} = 0$. The twin experiment truth has been obtained by integrating the model for $100$ time units. In particular, after a $20$ time units spin-up, the obtained solution in the time intervals $t \in [20, 40]$, $t \in [40, 60]$, $t \in [60, 80]$, $t \in [80, 100]$ has been referred as *truth_20*, *truth_40*, *truth_60*, *truth_80*.





### 4.1.2 Observations

Observations are generated for each truth period (*truth_20*, *truth_40*, *truth_60*, *truth_80*) by extracting from the state vector, the values of the even indexed variables (i.e., $x_2, x_4, ..., x_{62}$) every $\Delta t$ time units and adding to each observed variable a random number sampled from a standard normal distribution. Tests have been made for $\Delta t \in \{0.1, 0.15, 0.2, 0.25, 0.3\}$.

### 4.1.3 Filters setup

The same settings are used in both SEIK and GHOSH filters. The ensemble size was chosen among $7, 15, 31$ and $63$ members. The initial condition of the ensemble mean is chosen randomly around each *truth* initial condition, adding a random Gaussian noise with $0$ mean and a standard deviation of $5$. Initial conditions of the ensemble members are sampled with each filter's own sampling method (second order for SEIK, $h = 5$ for GHOSH), setting the prior error covariance to a diagonal matrix representing a standard deviation of $5$ in each variable.

Both filters have the same forgetting factor, chosen among $0.7, 0.75, 0.8, 0.85, 0.9, 0.95$ and $1$ (the last value describes a condition without inflation). The GHOSH filter is run without hybridization ($\mathbf{Q}_i = \mathbf{0}$) to keep GHOSH and SEIK as similar as possible. The order of the GHOSH filter was fixed at $h = 5$, with $s$ (i.e., the number of principal components approximated with order $h = 5$) as big as possible, depending on the ensemble size. The values adopted for $s$ are $2, 3, 4$ and $5$, respectively for $7, 15, 31$ and $63$ ensemble members.

### 4.1.4 Experiments design

100 twin experiments have been launched for each of the $4$ truth periods (*truth_20*, *truth_40*, *truth_60*, *truth_80*), 5 observation frequencies, 4 ensemble sizes and 7 forgetting factors. Each of the 100 twin experiments has its own set of random observations and initial conditions. Each experiment lasts 20 time units (as in the *truth* simulations), using the first half for spin-up. During the second half of the time interval, the root mean square error (RMSE) for assimilated and non-assimilated variables is separately measured before each assimilation.

All the code was written in python and executed in a Linux computer equipped with an Intel(R) Core(TM) i7-11800H @2.30GHz.

## 4.2 Three-dimensional marine biogeochemistry application

The GHOSH filter has been implemented in the OGSTM-BFM biogeochemical model system of the Mediterranean Sea at $\frac{1}{4}^o$ resolution. The GHOSH implementation features a weekly assimilation of surface chlorophyll from satellite ocean color (as in Teruzzi et al. (2018a)).

Consistently, an OGSTM-BFM model run without assimilation (control run) has been used as a reference to investigate the GHOSH filter sensitivity to some different parameterizations.





### 4.2.1 The OGSTM-BFM model

The biogeochemical model used in this study is the coupled transport-biogeochemical model OGSTM-BFM, which is the biogeochemical component of the Mediterranean Copernicus Marine Service (Salon et al. (2019)). The OGSTM transport model is a modified version of the OPA 8.1 transport model (Foujols et al. (2000)), which resolves the advection, the diffusion and the sinking terms of the biogeochemical variables (Lazzari et al. (2010)).

The biogeochemical BFM model describes the biogeochemical cycles of carbon, nitrogen, phosphorus and silicon through
the dissolved inorganic, particulate living organic and non-living organic compartments, and includes nine plankton functional types, as well as dissolved oxygen, pH and the carbonate system. The OGSTM-BFM model has been applied to study the chlorophyll and primary production (Lazzari et al. (2012); Teruzzi et al. (2018a)), nutrient cycles (Lazzari et al. (2016)), alkalinity, the carbon cycle and carbon fluxes (Cossarini et al. (2015); Melaku Canu et al. (2015)) and oxygen dynamics (Di Biagio et al. (2022)).

The OGSTM-BFM is forced by the output of an ocean general circulation model (i.e., NEMO, from the Copernicus Marine Mediterranean system in the present application; Salon et al. (2019)) that provides fields of physical quantities (i.e., velocity, temperature, salinity, diffusivity, and solar radiation and wind at the surface from a physical reanalysis) driving the transport processes, the kinetic rates of chemical reactions, and energy and matter exchanges at the air-sea interface.

The present setup (terrestrial and atmospheric inputs, boundary conditions) is described in Salon et al. (2019).

### 4.2.2 Observations

The assimilated observations are satellite chlorophyll maps obtained from the CMEMS multisensor product OCEANCOLOUR_MED_CHL_L3_REP_OBSERVATIONS_009_073 (Volpe et al. (2017)). The original data at 1 km resolution were checked for spikes, excluding values exceeding by two standard deviations the climatological mean and remapped on the model grid resolution.

Additionally, in situ climatological data for nitrate and phosphate, to be used for validation, is computed from a dataset that integrates the in situ EMODnet data collections (Buga et al. (2018)) and the datasets listed in Lazzari et al. (2016). The climatology is calculated for the 0-30 m layer on 9 sub-basins (Salon et al., 2019) and for 4 different seasons (January-March; April-June; July-September; October-December).

### 4.2.3 GHOSH setup

In the 3D simulations, GHOSH assimilated surface chlorophyll and update the whole state vector (i.e., 51 state variables) of the OGSTM-BFM model by using an ensemble of 16 members (i.e., an error subspace of dimension $r = 15$). Tests were carried out for each of three different maximum polynomial orders $h$ (and corresponding numbers $s$ of principal components approximated with such orders):

- $h = 2$, i.e., a second order of approximation on the whole error subspace ($s = 15$);



| Test name | Order ($h$) | Forgetting factor ($\rho$) | Forecast frequency |
|---|---|---|---|
| *ctrl* | | Free model | |
| *T_h2_rho1_d* | 2 | | |
| *T_h3_rho1_d* | 3 | 1 | |
| *T_h5_rho1_d* | 5 | | |
| *T_h2_rho0.9_d* | 2 | | daily |
| *T_h3_rho0.9_d* | 3 | 0.9 | |
| *T_h5_rho0.9_d* | 5 | | |
| *T_h2_rho0.8_d* | 2 | | |
| *T_h3_rho0.8_d* | 3 | 0.8 | |
| *T_h5_rho0.8_d* | 5 | | |
| *T_h2_rho1_w* | 2 | | |
| *T_h3_rho1_w* | 3 | 1 | |
| *T_h5_rho1_w* | 5 | | |
| *T_h2_rho0.7_w* | 2 | | weekly |
| *T_h3_rho0.7_w* | 3 | 0.7 | |
| *T_h5_rho0.7_w* | 5 | | |
| *T_h2_rho0.5_w* | 2 | | |
| *T_h3_rho0.5_w* | 3 | 0.5 | |
| *T_h5_rho0.5_w* | 5 | | |

**Table 1.** Experimental setup.

- $h = 3$, i.e., third order of approximation on the 8 most relevant components ($s = 8$) and second order on the remaining 7 principal directions;

- $h = 5$, i.e., fifth order of approximation on the top 3 principal components ($s = 3$), and second order on the rest (12 components).

The ensemble weights and $\mathbf{\Omega}^h$ were computed (usually during the *initialization* phase) by solving system (1). In the case with $h = 2$, the solution is easily obtained starting from the constant vector $v$ of the square root of the weights and adding orthonormal column vectors in order to form an orthogonal matrix (see equation (A10) in Appendix A).





In the case with $h = 3$, a solution with constant weights ($v_m = \frac{1}{4}$ for each $m \in \{1, \dots, 16\}$) was chosen, namely,

$$\boldsymbol{\Omega}^{h=3} = \begin{pmatrix} \frac{1}{\sqrt{2}} & -\frac{1}{\sqrt{2}} & 0 & 0 & \cdots & 0 & 0 & 0 & 0 \\ 0 & 0 & \frac{1}{\sqrt{2}} & -\frac{1}{\sqrt{2}} & \cdots & 0 & 0 & 0 & 0 \\ \vdots & \vdots & \vdots & \vdots & \ddots & \vdots & \vdots & \vdots & \vdots \\ 0 & 0 & 0 & 0 & \cdots & \frac{1}{\sqrt{2}} & -\frac{1}{\sqrt{2}} & 0 & 0 \\ 0 & 0 & 0 & 0 & \cdots & 0 & 0 & \frac{1}{\sqrt{2}} & -\frac{1}{\sqrt{2}} \end{pmatrix}^{T}. \tag{45}$$

In the case with $h = 5$, the chosen solution was

$$\boldsymbol{v} = \left( \begin{array}{ccccccccc} \frac{1}{\sqrt{5}} & \frac{1}{\sqrt{5}} & \frac{1}{2\sqrt{5}} & \frac{1}{2\sqrt{6}} & \frac{1}{2\sqrt{6}} & \frac{1}{2\sqrt{6}} & \frac{1}{2\sqrt{6}} & \frac{1}{2\sqrt{6}} & \frac{1}{2\sqrt{6}} & \cdots \end{array} \right.$$

$$\left. \begin{array}{ccccccc} \cdots & \frac{1}{2\sqrt{5}} & \frac{1}{2\sqrt{6}} & \frac{1}{2\sqrt{6}} & \frac{1}{2\sqrt{6}} & \frac{1}{2\sqrt{6}} & \frac{1}{2\sqrt{6}} & \frac{1}{2\sqrt{6}} \end{array} \right)^{T} \tag{46}$$

and

$$\boldsymbol{\Omega}^{h=5} = \begin{pmatrix} 0 & 0 & \frac{1}{2} & \frac{1}{2\sqrt{6}} & \frac{1}{2\sqrt{6}} & \frac{1}{2\sqrt{6}} & \frac{1}{2\sqrt{6}} & \frac{1}{2\sqrt{6}} & \frac{1}{2\sqrt{6}} & \cdots \\ 0 & 0 & 0 & \frac{1}{\sqrt{6}} & \frac{1}{2\sqrt{6}} & \frac{1}{2\sqrt{6}} & -\frac{1}{\sqrt{6}} & -\frac{1}{2\sqrt{6}} & -\frac{1}{2\sqrt{6}} & \cdots \\ 0 & 0 & 0 & 0 & \frac{1}{2\sqrt{2}} & -\frac{1}{2\sqrt{2}} & 0 & \frac{1}{2\sqrt{2}} & -\frac{1}{2\sqrt{2}} & \cdots \end{pmatrix}$$

$$\begin{array}{ccccccc} \cdots & -\frac{1}{2} & -\frac{1}{2\sqrt{6}} & -\frac{1}{2\sqrt{6}} & -\frac{1}{2\sqrt{6}} & -\frac{1}{2\sqrt{6}} & -\frac{1}{2\sqrt{6}} & -\frac{1}{2\sqrt{6}} \\ \cdots & 0 & \frac{1}{\sqrt{6}} & \frac{1}{2\sqrt{6}} & \frac{1}{2\sqrt{6}} & -\frac{1}{\sqrt{6}} & -\frac{1}{2\sqrt{6}} & -\frac{1}{2\sqrt{6}} \\ \cdots & 0 & 0 & \frac{1}{2\sqrt{2}} & -\frac{1}{2\sqrt{2}} & 0 & \frac{1}{2\sqrt{2}} & -\frac{1}{2\sqrt{2}} \end{array} \Bigg)^{T}. \tag{47}$$

The scaling matrix $\boldsymbol{\Lambda}_i$ (equation (7)) is a diagonal matrix representing the error variance (i.e., the diagonal of $\mathbf{P}_i^a$), and it is computed as the squared Euclidean norms of the rows of $\mathbf{L}_i^a$. The $\boldsymbol{\Lambda}_i$ matrix has been chosen valuing simplicity and generality over skill performance. The implemented scaling matrix is probably a suitable choice for many systems, since it implies that the PCA measures the Pearson correlation and that it is not affected by the different amplitudes of the variable standard deviations.

The simulations cover the period 1 January 2013 - 1 January 2014, with daily or weekly forecast frequency (1).

The ensemble covariance was inflated as in equation (32). Tests were conducted with different forgetting factor values, namely, $\rho \in \{1, 0.9, 0.8\}$ for daily forecasts and $\rho \in \{1, 0.7\}$ for weekly forecasts.

The localized version of the filter used cylindrical and circular patches for $\mathcal{L}_p$ and $\mathcal{L}_p^{\mathcal{H}}$, using a 90 km radius (similarly to values adopted by other EnKF applications, e.g., Hu et al. (2012); Ciavatta et al. (2016)) centred at the point $p$. The delocalization operator $\mathcal{D}$ builds the global array by taking the values in the central cell of each localized array (Section 3.2). However, since the change in basis in the GHOSH algorithm was continuous (as noted in Section 3.2.1), more demanding localization strategies, such as averaging over the weighted values of each cell local array, would not introduce relevant improvements.

Given the non-Gaussianity of biogeochemical tracer concentrations, the GHOSH filter was applied to the log-transformed biogeochemical variables (Ciavatta et al. (2016); Pradhan et al. (2019); Song et al. (2016); Ford (2020)). To avoid instabilities





due to large negative values of logarithms of very small concentrations, a cut-off was applied to values lower than the threshold of $10^{-4}$. This solution can be seen as a localization that dynamically removes unwanted correlations, which can typically occur at deeper layers, where some tracers (such as, for example, phytoplankton biomass and chlorophyll) tend to very low concentrations.

The mean initial condition $\boldsymbol{x}_0^f$ and the starting ensemble $\mathbf{X}_0^f$ were produced based on a 10-year hindcast simulation of the OGSTM-BFM model without assimilation. For each year, 31 days centred on the 1st of January were log-transformed, after removing values smaller than $10^{-4}$, for a total of 310 simulated multivariate states. The mean of the multivariate states was used as $\boldsymbol{x}_0^f$, while a PCA of the ensemble of normalized states provided the 15 (i.e., the dimension of the error subspace $r$) most relevant components, which are used to build the base matrix $\mathbf{L}_0^f$. Then, the GHOSH sampling method has been used to
extract the 16 starting ensemble members, as in Section 3.1.5.

The local model error covariance matrix $\mathbf{Q}_i^p$ has been assumed diagonal and invariant in time. The variances in $\mathbf{Q}_i^p$ are assumed inversely proportional to the height of the cells. Given that the resolution of the present model setup is not high enough to adequately resolve coastal processes, assimilating observations in coastal areas could lead to instabilities. Therefore, the model error variance in the coastal areas was reduced by applying a 4th-degree polynomial smoothing function and setting
the covariance values at zero in the grid cells closest to the coast. More precisely, the uncertainty starts to decrease where sea depth is 200 metres and reaches zero in areas shallower than 50 metres. The local observation error covariance matrix $\mathbf{R}_i^p$ was assumed diagonal and constant in time. Using the same 4th-degree polynomial smoothing applied for $\mathbf{Q}_i^p$, the inverse of the $\mathbf{R}_i^p$ variances was smoothed to zero in the coastal areas (i.e., the $\mathbf{R}_i^p$ variances increase in the coastal areas oppositely to $\mathbf{Q}_i^p$). Moreover, the matrix $(\mathbf{R}_i^p)^{-1}$ was weighted with the polynomial function in equation (41) to apply the local version of the
filter.

A tuning of the model and observation error variance may be necessary to produce an effective assimilation when system errors are not fully known and parametrized (e.g., Cossarini et al. (2019); Ford and Barciela (2017); Janjić et al. (2018); Mattern et al. (2017); Tandeo et al. (2020)). The actual values of model and observations error variances were obtained testing different values and choosing those that provided effective assimilation increments without introducing biogeochemical variables
degradation.

In total, 18 assimilation experiments were run, in addition to the control run, as summarized in Tab. 1. Overall, the experimental design features a number of different forgetting factors $\rho$, 3 values of the approximation order $h$ (from 2 to 5) and 2 different time intervals between two *forecast* phases (1 and 7 days) to test the sensitivity of the assimilation scheme.

The tests were carried out in the GALILEO cluster at the Italian HPC centre CINECA, mounting 2 Intel Broadwell CPU
(Intel(R) Xeon(TM) E5-2697 v4 @2.3GHz, 18 cores each) per each node. The parallel Fortran 2003 implementation of the GHOSH algorithm ran on 46 nodes (corresponding to 1656 cores).





## 5 Results

### 5.1 Twin Experiment

Figure 3 (a and b) reports the evolution of the first 2 variables (one assimilated and one non-assimilated) of the Lorenz96 model
for the two filters (GHOSH and SEIK). The selected simulation out of 56000 experiments shows that GHOSH filter evolution
is closer to truth evolution (black line in Fig. 3) than SEIK. As expected, the uncertainty of variable 2 (shaded areas) of both
filters is lower than in variable 1 because of the assimilation of observations in even variables. The RMSE computed at every
time step over the two sets of variables (assimilated and non-assimilated, Fig. 3) shows that GHOSH outperforms SEIK after
the spin-up time.

In order to be statistically reliable, the RMSE of the 56000 experiments under different conditions have been computed and
reported in aggregated form in Fig. 4. Each small square reports the RMSE value computed over a set of 400 experiments,
which includes, for each of the four available truths, 100 experiments with different random observations and initial conditions.
The first 2 lines of color maps refer to the SEIK filter observed and unobserved variables. In each color map, the time interval
between assimilated observations increases from left to right, while the different forgetting factor values decrease from top to
bottom. The first line in each color map, labelled "*best*", represents the best result obtained between all the tested forgetting
factors.

The GHOSH experiments are summarized in the same way in the 3rd and 4th lines of Fig. 4, while the last 2 lines show the
ratio between GHOSH and SEIK RMSE values, with red color indicating RMSE reduction of GHOSH with respect to SEIK.

The column corresponding to the smallest ensemble size (i.e., 7 members) is not shown, since in this case SEIK and GHOSH
behave very similarly, showing very poor performances, due to lack of capability of describing the system complexity with an
overly small ensemble size. In these conditions the GHOSH showed a very small improvement with respect to SEIK (0-3%).

Looking at the last 2 rows of Fig. 4, the GHOSH filter outperforms the SEIK filter in most conditions, up to a three times
RMSE reduction. Furthermore GHOSH proves to be always at least as good as SEIK. The largest improvements (dark red,
RMSE ratio 0.31) occur when: i) inflation is low (forgetting factor near to 1); ii) large amount of information is injected into
the assimilation scheme (i.e., high frequency observations, left side of Fig. 4); iii) the filter can take into account a high dimen-
sional error subspace (i.e., the ensemble size is large).

On the other hand, the comparison of each filter tuned with its best forgetting factor (the "*best*" top line in each color map)
clearly shows that the GHOSH filter has considerably better performances than the SEIK filter (up to a RMSE ratio of 0.43)
with a moderate number of ensemble members (31). In case of maximum ensemble size the improvement is still relevant but
less intense (up to 0.9 RMSE ratio). The improvement is comparably small, up to 7%, in the case of small ensemble size (15
members).

In general, it is evident that the GHOSH filter, compared to SEIK, is less dependent on inflation, reaching its best performance
with a forgetting factor closer to 1.

The skill of both the filters is closely related with the observations frequency: the shorter the time between observations, the
higher the accuracy.



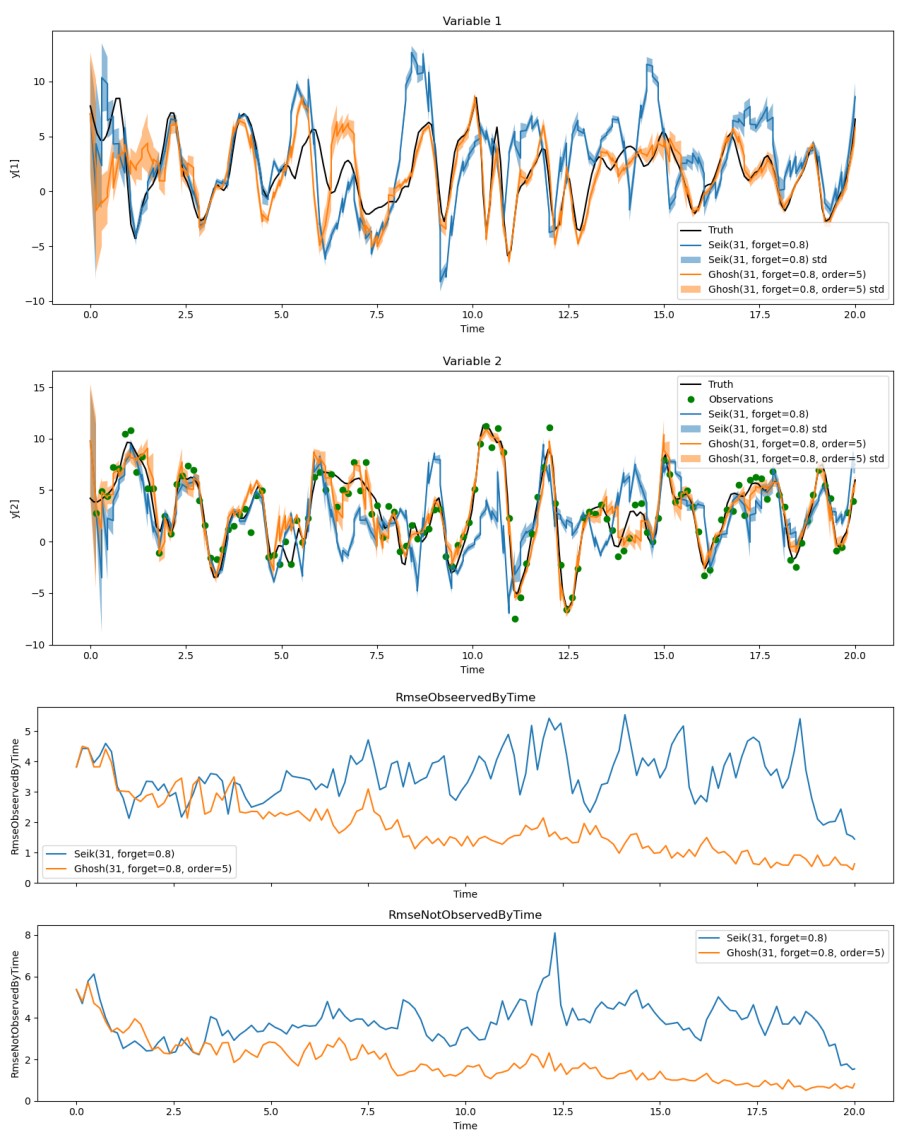

**Figure 3.** Results from one twin experiment example: *truth* (black line) is *truth_60*, time between observation (green dots) is 0.15, forgetting factor is 0.8, ensemble size is 31. Shady areas around blue (SEIK) and organge (GHOSH) lines represent the ensemble standard deviation. From top to bottom: time evolution of the first variable (non-assimilated); time evolution of the second variable (assimilated); RMSE along time based on all the assimilated variables; RMSE along time based on all the non-assimilated variables.





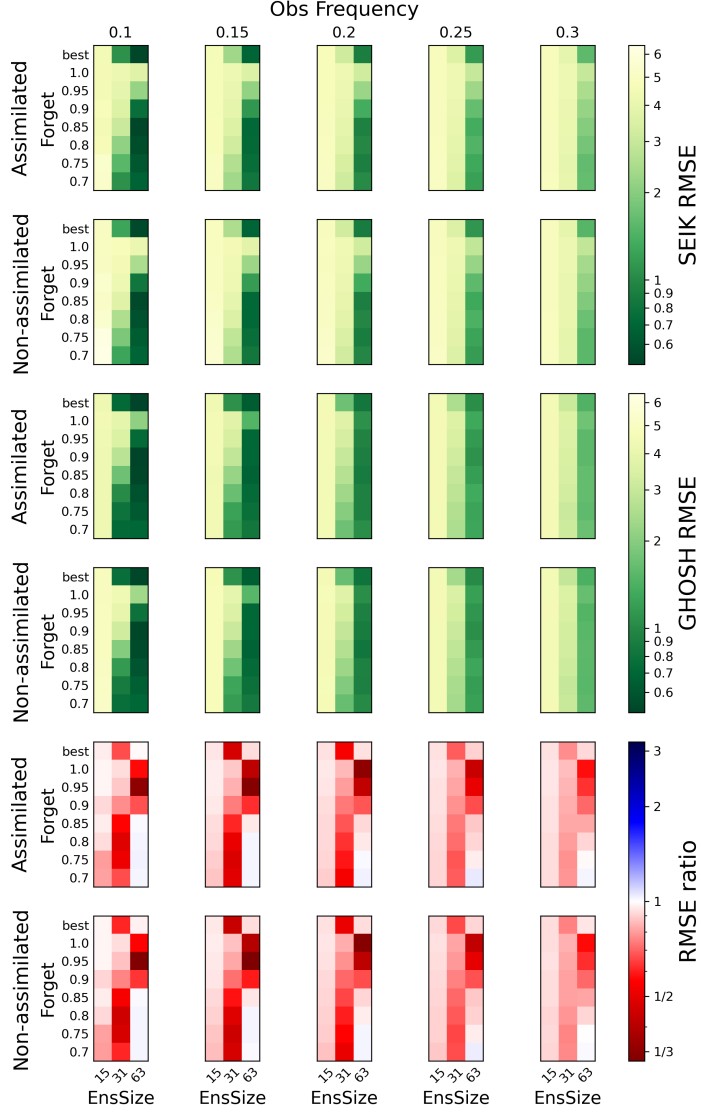

**Figure 4.** Result summary of twin experiments: each square in the color maps represents the aggregated results of 400 twin experiments, changing *truth*, observations and initial conditions. The results are summarized with color maps aggregated in six different rows, from top to bottom: SEIK RMSE of assimilated variables, SEIK RMSE of non-assimilated variables, GHOSH RMSE of assimilated variables, GHOSH RMSE of non-assimilated variables, the ratio of GHOSH RMSE over SEIK RMSE of assimilated variables, the ratio of GHOSH RMSE over SEIK RMSE of non-assimilated variables (red color implies that GHOSH is better than SEIK). Each column of color maps has different observation frequency, with the numbers on the top indicating the time elapsed between each observation/assimilation. Each color map shows different forgetting factors (Forget, along the $y$ axis) and ensemble sizes (EnsSize, along the $x$ axis).





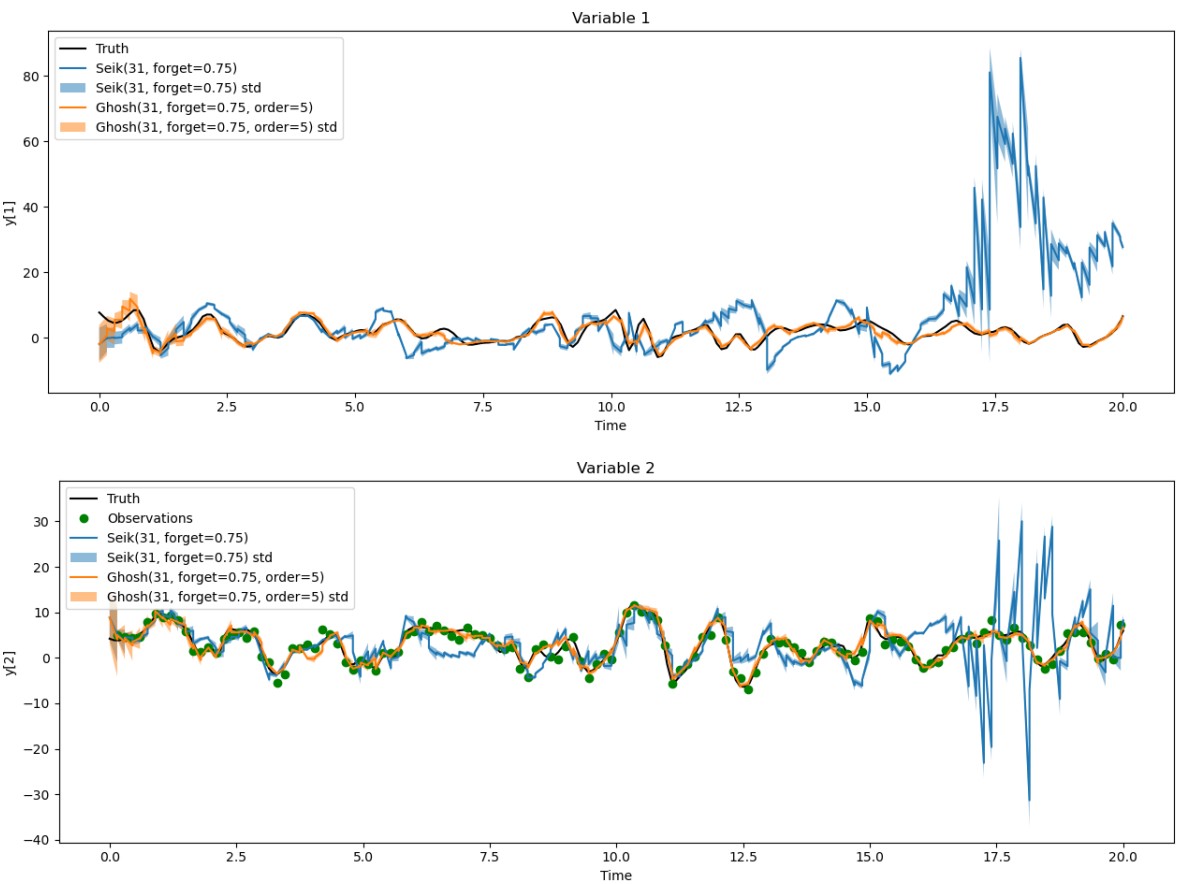

**Figure 5.** Example of a single twin experiment, with SEIK (blue line) diverging behaviour: *truth* (black line) is *truth_60*, time between observations (green dots) is 0.15, forgetting factor is 0.75, ensemble size is 31. Values over time of a non-assimilated (top) and an assimilated (bottom) variable.

Finally, as expected, for both the filters the skill is better on assimilated variables than non-assimilated ones but, quite interestingly, the RMSE reductions of GHOSH with respect to SEIK are slightly larger in the unobserved variables (up to 0.06 RMSE ratio improvement, from 0.67 for assimilated to 0.61 for non-assimilated variables).

From the computational point of view, the whole experiment set needed around 7 computational hours. Since the compu-515 tational cost is dominated by the integration of the model, the time to solution of the two tested filters should be nearly the same, however, unexpectedly, the average time to solution (averaged every 100 twin experiments) of the SEIK filter sometimes is longer than the time to solution of the GHOSH filter, up to twice the time, depending on some settings and random factors. This difference in computational time is more evident and occurs more often when the forgetting factor is lower (i.e., when





| Test name | Order | Chlorophyll | Phosphate | Nitrate |
| :---: | :---: | :---: | :---: | :---: |
| | $(h)$ | RMSD $(mg/m^3)$ | RMSD $(mmol/m^3)$ | RMSD $(mmol/m^3)$ |
| *ctrl* | no | 0.084 | 0.036 | 0.99 |
| *T_h2_rho1_d* | 2 | 0.068 | 0.035 | 0.93 |
| *T_h3_rho1_d* | 3 | 0.068 | 0.035 | 0.85 |
| *T_h5_rho1_d* | 5 | 0.068 | 0.035 | 0.83 |
| *T_h2_rho0.9_d* | 2 | 0.059 | 0.033 | 1.50 |
| *T_h3_rho0.9_d* | 3 | 0.060 | 0.033 | 1.14 |
| **T_h5_rho0.9_d** | **5** | **0.059** | **0.033** | **0.83** |
| *T_h2_rho0.8_d* | 2 | 0.059 | 0.033 | 1.96 |
| *T_h3_rho0.8_d* | 3 | 0.058 | 0.032 | 2.15 |
| *T_h5_rho0.8_d* | 5 | 0.058 | 0.032 | 1.76 |
| *T_h2_rho1_w* | 2 | 0.071 | 0.035 | 0.94 |
| *T_h3_rho1_w* | 3 | 0.071 | 0.035 | 0.89 |
| *T_h5_rho1_w* | 5 | 0.071 | 0.035 | 0.91 |
| *T_h2_rho0.7_w* | 2 | 0.063 | 0.034 | 0.94 |
| *T_h3_rho0.7_w* | 3 | 0.063 | 0.035 | 0.93 |
| *T_h5_rho0.7_w* | 5 | 0.063 | 0.034 | 0.91 |
| *T_h2_rho0.5_w* | 2 | 0.060 | 0.033 | 1.35 |
| *T_h3_rho0.5_w* | 3 | 0.060 | 0.032 | 1.02 |
| *T_h5_rho0.5_w* | 5 | 0.060 | 0.033 | 0.91 |

**Table 2.** Results: summary of the root mean square difference (RMSD) indicators for each test. Best GHOSH configuration in bold.

the inflation is more pronounced). A possible explanation comes from the model integration algorithm used in the SciPy's

*solve_ivp* routine: the integration time step is not constant and depends on the stability of the equations (stiffness). Which means that, if the filter is less accurate, the output can diverge from the expected trajectory in areas where the equations are stiffer, and the integration needs more time steps to converge rising the time to solution. This is supported by the fact that the SEIK filter can show a diverging behaviour when the forgetting factor is too low (Fig. 5 shows an example). Remarkably, we did not observe any diverging behaviour in the GHOSH filter runs.

**5.2 Three-dimensional realistic complexity marine biogeochemistry application**

The GHOSH capability to reproduce expected dynamics in a complex and real geophysical application has been evaluated by looking at: the skill performance of forecast of the assimilated variable (chlorophyll) and of the non-assimilated variables (nutrients) and the quality of the system state uncertainty estimation.





Furthermore, the GHOSH performances have been evaluated varying the order of approximation $h$. Table 2 shows that the
performance of the assimilated variable is substantially unaffected by the GHOSH polynomial order, whilst $h$ demonstrates a
relevant impact on a non-assimilated variable, namely the nitrate. In particular, when the order $h$ is 5, nitrate root mean square
difference (RMSD) between GHOSH and independent observations improves up to $81\%$ with respect to order $h = 2$ (scored
in the set of tests with $\rho = 0.9$ and daily forecasts). Every $h = 5$ test has a smaller RMSD than every corresponding $h = 2$ and,
in general, data clearly show that increasing $h$ implies an improvement in RMSD. In fact, the nitrate's RMSD values, centred
and normalized over their standard deviation, present a strong negative correlation with $h$ (Pearson's correlation coefficient is
$-0.80$).

  Among the 18 assimilation experiments, *T_h5_rho0.9_d* proved to have the best GHOSH configuration (Tab. 2). The results
from this run are discussed in detail in the following sections, through a qualitative and quantitative comparison with the control
run (i.e., a hindcast run with no assimilation).

**5.2.1 Surface chlorophyll**

The surface chlorophyll estimated by the free run and the GHOSH filter has been compared with satellite data from January
to December 2013. A detailed analysis of a specific relevant event (winter bloom, 19 March 2013) is shown in Fig. 6). The
RMSD time series is shown in Fig. 7.

  The March map describes a typical winter bloom in the western Mediterranean, with the northernmost part still under the
convective phase (i.e., blue hole surrounded by high chlorophyll concentration patches, Fig. 6a). Additionally, high chlorophyll
levels are present along the Algerian Current as a result of the active mesoscale dynamics, whereas the eastern Mediterranean
Sea has a much lower chlorophyll concentration due to the lower intensity of the winter vertical mixing (Fig. 6a).
While the free run overestimates the surface chlorophyll in the whole Mediterranean Sea (Fig. 6b), the pre-assimilation picture
of GHOSH run shows a more consistent condition (Fig. 6c), which is further improved by the assimilation (Fig. 6d).
Time series of RMSD computed between satellite and model before the assimilation are plotted in Fig. 7. Since the amount of
surface chlorophyll can vary considerably during the year, the relative root mean square difference (RRMSD) is also provided
(i.e., RMSD over concentration), as a normalized proxy of the skill. Figure 7 shows that the GHOSH assimilation produces an
improvement in the assimilated variable (i.e., surface chlorophyll) with respect to the control run. The RMSD of the control run
follows a seasonal cycle, with maximum values in late winter-early spring, corresponding to the bloom period, and minimum
values during summer when the surface layer is mostly nutrient depleted and chlorophyll is at its lowest level.
The winter-spring RMSD is significantly reduced by the GHOSH filter whereas the RMSD reduction results fairly small during
summer, due to a general smaller amount of surface chlorophyll in that season. In fact, when the RRMSD is considered, the
summer improvement due to assimilation is more detectable.
  As a proxy of the overall performance of each 1-year experiment, the RMSD between the simulation and observation, averaged
in space and time, is summarized in Tab. 2: the chlorophyll RMSD for *ctrl* (control run) is 0.084, while it is 0.055 for the
GHOSH test, resulting in a $53\%$ improvement.



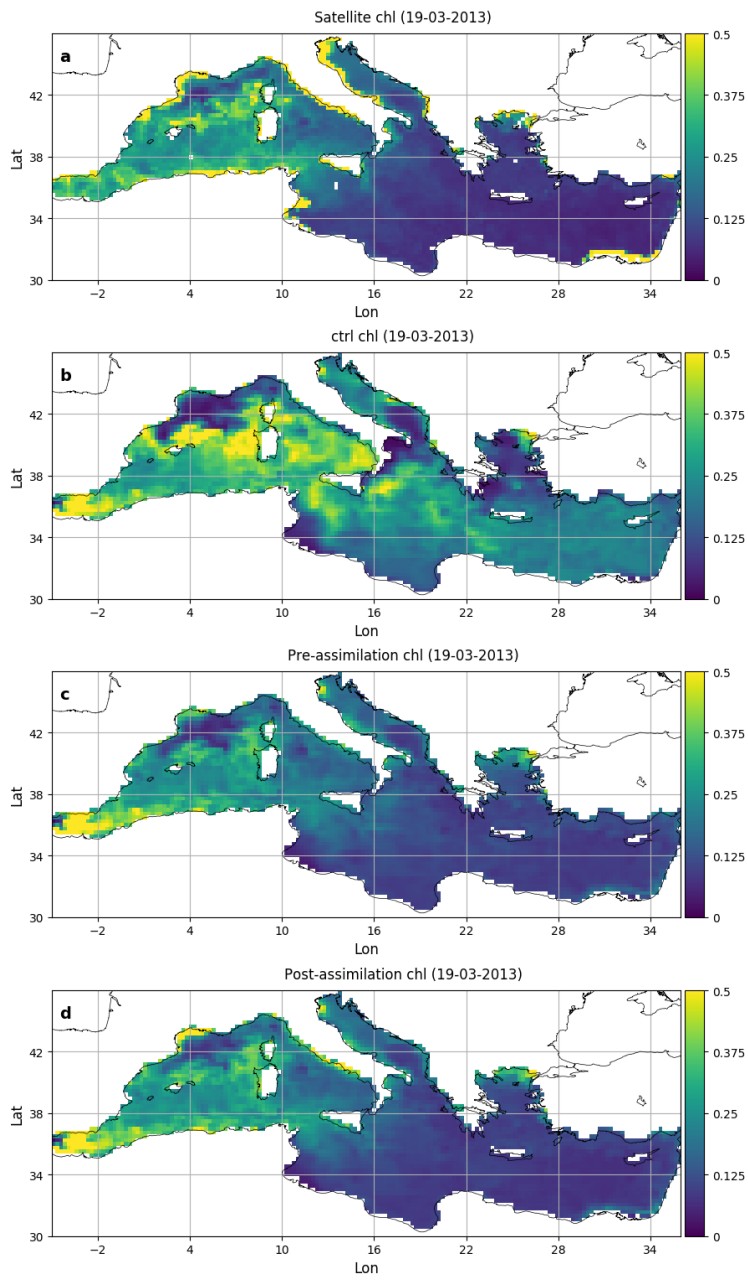

**Figure 6.** Surface chlorophyll ($mg/m^3$) comparison: satellite, control run and data assimilation (before and after assimilation) on 19.03.2013.





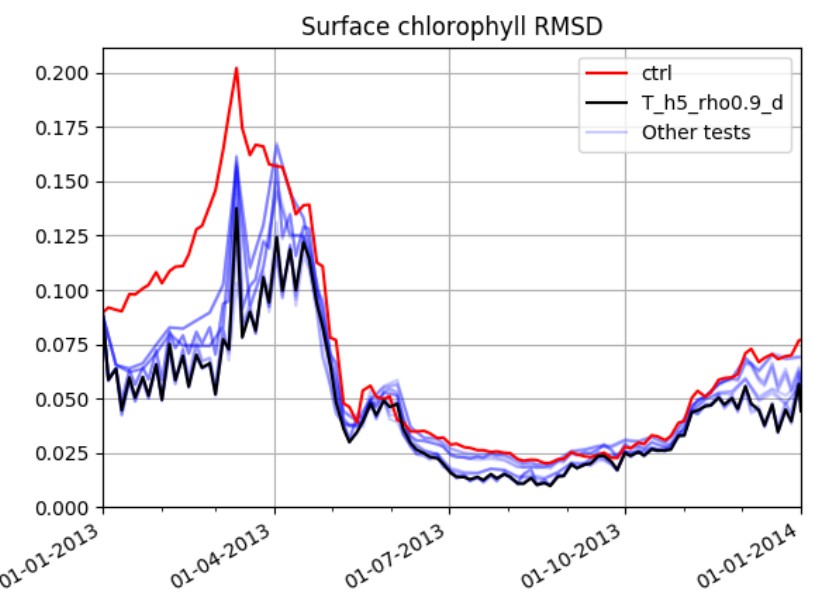

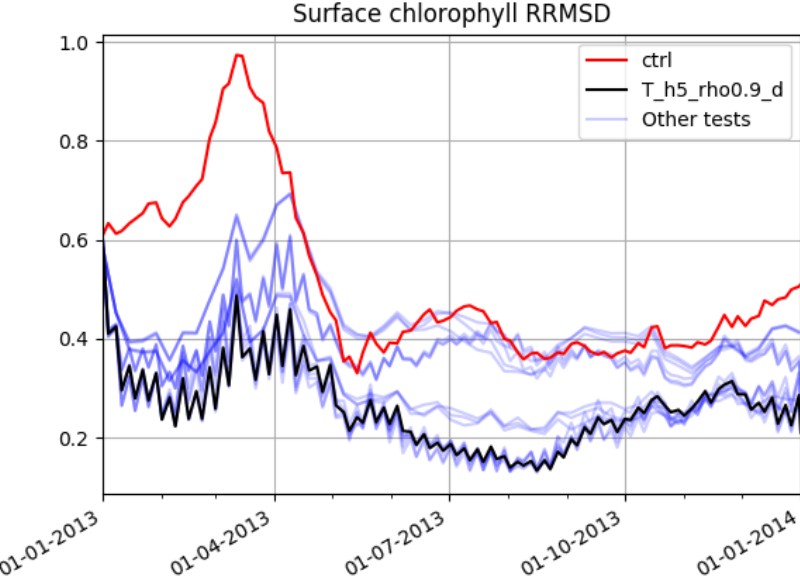

**Figure 7.** Surface chlorophyll RMSD ($mg/m^3$) and RRMSD over time.

### 5.2.2 Nutrients

The effect of DA on non-assimilated variables has been assessed using independent observations to validate phosphate and nitrate (Tab. 2), which can both act as limiting nutrient for phytoplankton growth in the Mediterranean Sea (Lazzari et al.

(2016)).



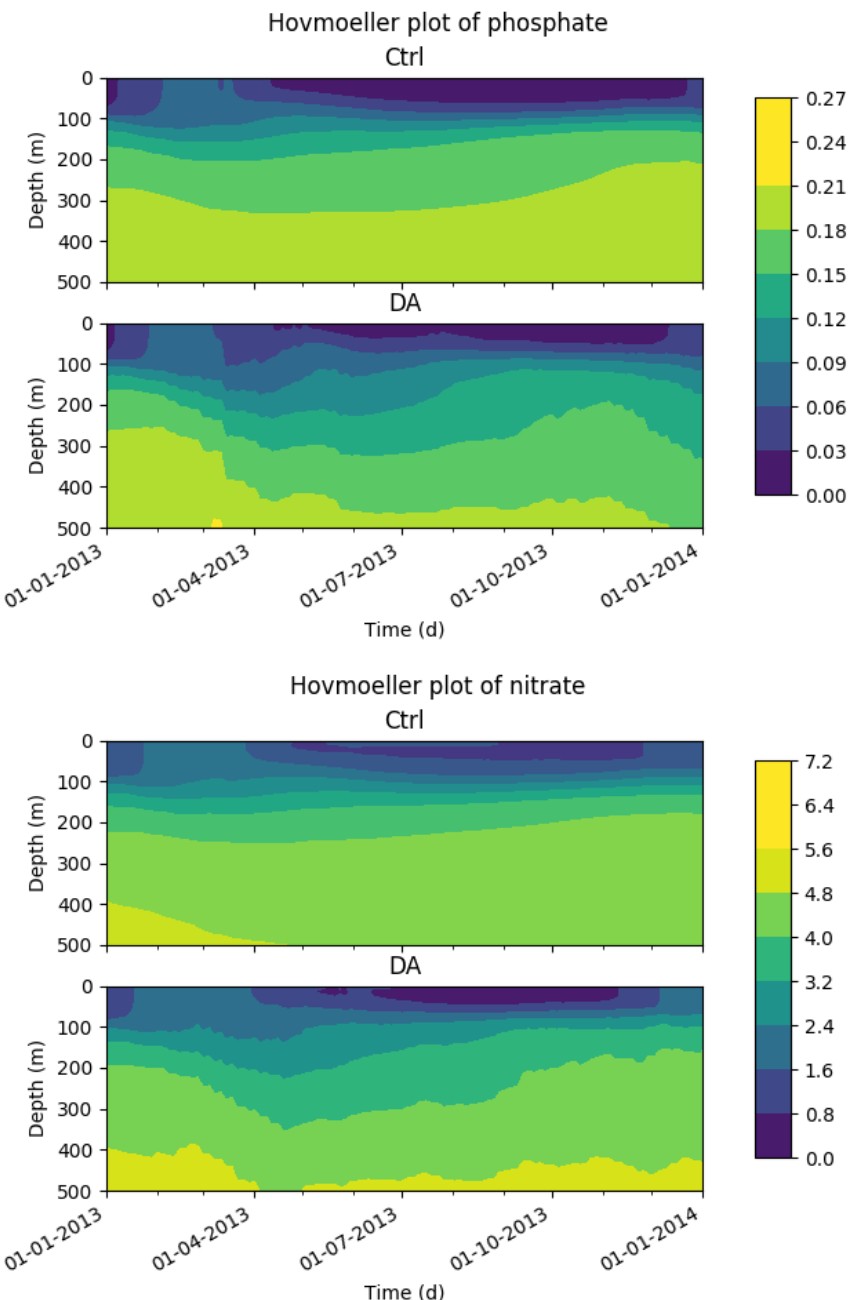

**Figure 8.** Hovmöller diagrams of nutrients (phosphate and nitrate, $mmol/m^3$): comparison between the control run *ctrl* (up) and data assimilation test *T_h5_rho0.9_d* (bottom)





The Hovmöller diagrams of phosphate and nitrate in the DA and the control runs show the effects of the assimilation on the two nutrient compartments (Fig. 8). Both nitrate and phosphate show lower concentrations in the 150–400 m layer from April in the simulation with satellite assimilation, with highest impact at the beginning of April. At the surface (0–100 m layer), the assimilation slightly enhances the phosphate concentrations, particularly between April and July, while it reduces the nitrate

concentrations. However, these effects do not degrade the typical vertical distribution of nutrients in the Mediterranean Sea, and the seasonal sequence of late-spring/winter mixed conditions and of summer stratification is preserved.

Interestingly, the Hovmöller diagrams of Fig. 8 show that the DA is able to correct the profiles of the nutrients down to 300–400 m by changing the shape of the nutricline (i.e., deepening and creating a gentler shape) during the month of the deep chlorophyll maximum formation (from April to June) and by re-setting the nutricline to a shallower level after summer.

### 575 5.2.3 Uncertainty estimation

In the GHOSH filter, a weighted ensemble is used to estimate the uncertainty of the forecast and analysis states (equation (12)). The weighted ensemble standard deviation is used to quantify the uncertainty, whereas a comparison of the standard deviations before and after assimilation is calculated to evaluate the uncertainty reduction produced by the assimilation. Moreover, the ensemble correlation matrix is analyzed to investigate the relationships between variables. The standard deviation and correlation

are shown for the assimilated variable and one of the nutrients during a bloom event in winter.

During the winter event (March), the spatial patterns of chlorophyll and phosphate standard deviations at the surface (second row of Fig. 9) reflect those of their concentration maps (first row of Fig. 9): areas with high surface concentrations in the western Mediterranean Sea are associated with the highest values of the ensemble standard deviation given the high variability of the surface bloom dynamics. The map also shows an area with low chlorophyll and high phosphate concentration in the

northwestern Mediterranean where winter convection is still active and standard deviation is low.

As an effect of the assimilation, the areas with the highest standard deviation for both variables in the western Mediterranean Sea decrease, proving the effectiveness of the surface assimilation in constraining the surface phytoplankton bloom dynamics. Moreover, a relevant reduction in the standard deviation, in relative terms, is also observed in the less productive eastern Mediterranean Sea.

On average, the assimilation reduces the standard deviation at the surface by $33\%$ and $22\%$ for chlorophyll and phosphate, respectively.

A few spots of high standard deviations remain only in some coastal areas in the western Mediterranean Sea for both variables. The system has been tuned to focus on open sea and the assimilation in shallow waters is less effective due to imposed higher observation errors in the coastal areas (see Section 4.2.3 for details).

### 595 5.2.4 Three-dimensional Feasibility and computational cost

The feasibility of the GHOSH scheme in a realistic 3D operational system has been confirmed by the experiments: each 1-year assimilation test, consisting of 16 ensemble members, needed approximately 4 hours to be computed in the CINECA's GALILEO super computer, running in parallel on 46 computational nodes (1656 cores).



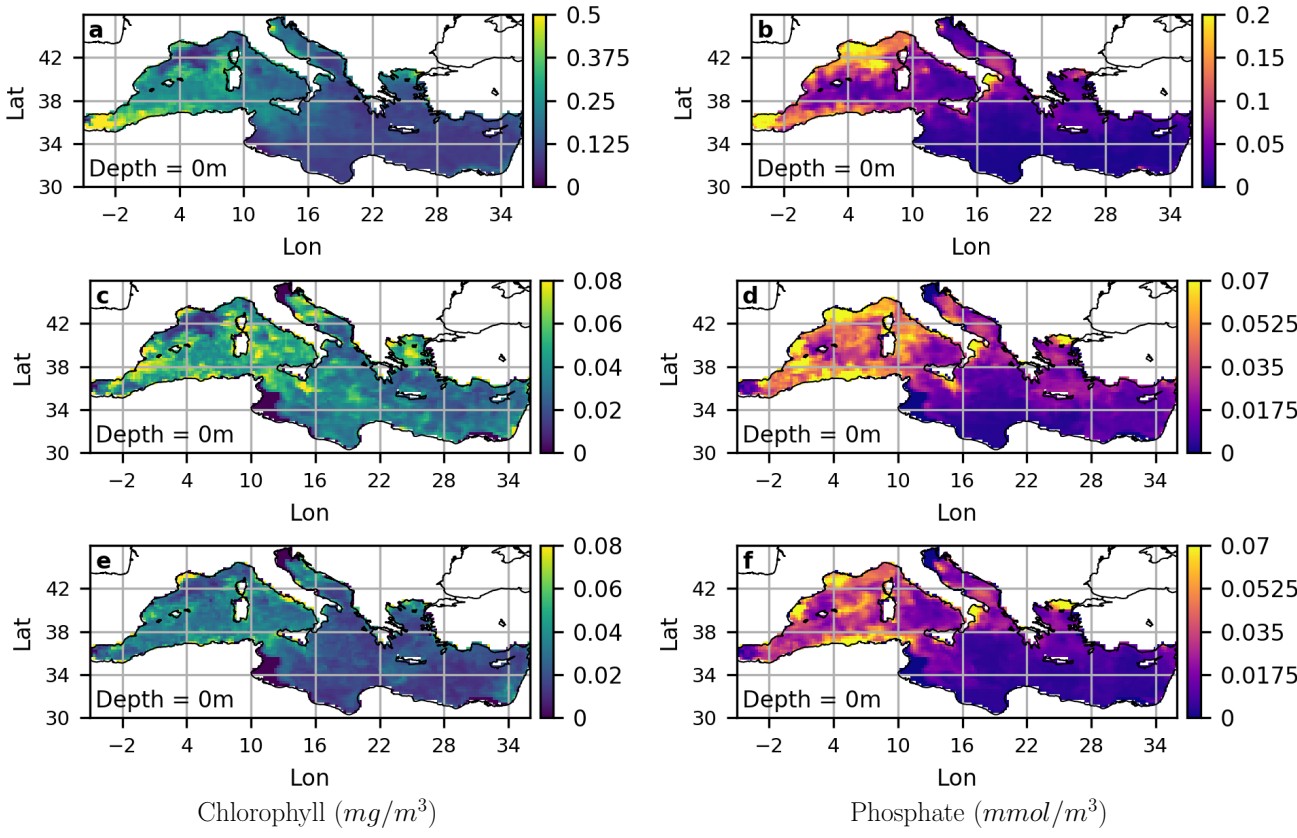

**Figure 9.** Maps of chlorophyll and phosphate (a and b) and maps of their uncertainty (ensemble standard deviation) before assimilation (c and d) and after assimilation (e and f) for the day 19.03.2013.

The computational cost of the GHOSH filter itself is negligible compared with the model integration time and the I/O opera-

tions, being around $1\%$ of the total computational time.

## 6   Discussion

The name "Gauss-Hermite high-order sampling hybrid filter" was chosen to indicate the two main features of this novel filter. First, it is a hybrid filter, in the sense that the error covariance matrix is obtained by averaging the ensemble covariance and a background error covariance matrix, as described in Carrassi et al. (2018). Second, it exploits a novel sampling method based

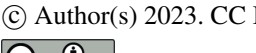



on a Gauss-Hermite-like quadrature rule to reach an arbitrary high (depending on the ensemble size) polynomial order of approximation.

The latter feature introduces, to the best of our knowledge, a completely new class of DA algorithms with an order higher than 2. The advantage of moving to a higher order of approximation has previously been documented in the literature, in particular Nerger et al. (2005) compares order 1 algorithms (e.g., SEEK Pham et al. (1998)) and order 2 algorithms (e.g., SEIK

Pham (1996)). Results of Section 5 confirm the expected benefits of the high order sampling adopted in the GHOSH filter. In particular, in the Lorenz96 idealized and controlled conditions, the novel method outperforms (up to 3 times better RMSE) a typical second order method like SEIK and features higher stability and better performances.

At the same time, in a realistic 3D high dimensional parallel implementation with assimilation of real satellite data, the GHOSH filter proved (in Section 5.2) to be a reliable data assimilation scheme with $30\%$ RMSD reduction for the assimilated variable.

Moreover, the GHOSH assimilation does not degrade non-assimilated variables, actually improving their RMSDs with respect to independent observations (figures 7 and 8). Consistently, the GHOSH assimilation results allowed catching some relevant system dynamics (Fig. 9). Further, the sensitivity analysis performed in the 3D application underlines the role of the high order sampling in improving performances in a non-assimilated variable (Tab. 2).

Thanks to the large number of settings tested in the Lorenz96 twin experiment, it is possible to suggest the conditions

where the GHOSH filter considerably increases the assimilation skill and reliability. Concerning the ensemble size, the larger improvements occur when the dimension of the error subspace spanned by the ensemble is big enough to be representative. In other words, if the number of ensemble members is too small, the assimilation simply fails independently from the assimilation algorithm. Moreover, it has been shown that the GHOSH increased accuracy reduces instabilities and numerical divergence (Fig. 5). This aspect is consistent with the fact that a higher order of approximation helps to manage non-linearities (Nerger

et al. (2005)). GHOSH filter showed a lower need for inflation with respect to SEIK, removing one of the instability causes observed in the twin experiment (Section 5.1).

Together with the GHOSH's lower need of inflation, the hybridization adopted in the 3D application prevented the "collapse into the mean" effect (not shown), which often affects ensemble algorithms (Anderson (2007)). Indeed, the hybridization adopted in each *forecast* phase in the GHOSH 3D application has the beneficial effect of maintaining a reasonable spread, even

without the forgetting factor (i.e., $\rho = 1$).

The results obtained coupling the GHOSH filter to the OGSTM-BFM model show a major improvement in the nitrate performance when increasing the approximation order $h$. However, chlorophyll and phosphate RMSDs are not substantially affected by $h$. Consistently with the twin experiments results (Section 5.1), this behaviour may depend on the relatively low number of ensemble members (16, $r = 15$), which imply that only a limited number of principal components ($s = 3$) takes

advantage of the higher order $h = 5$. Indeed, the negligible effects of the higher approximation order on chlorophyll and phosphate could be motivated by the fact that, with the normalization rules chosen in this experiment (i.e., $\Lambda_i$), they possibly contribute little to the few most relevant components of the state uncertainty. A larger number of ensemble members (i.e., bigger $r$) would allow a larger $s$-subspace, probably extending the high-order effects on further state variables. This view is





consistent with the generally observed benefits of a larger ensemble in EnKF-based data assimilation (Edwards et al. (2015);
Nerger et al. (2005); Sacher and Bartello (2008)).

The scaling matrix $\mathbf{\Lambda}_i$ used in the sampling procedure plays a relevant role in the filter, since $\mathbf{\Lambda}_i$ defines how variables are compared to each others in the computation of the state error most relevant components (Equation (7)). In the Lorenz96 case, it is straightforward to set $\mathbf{\Lambda}_i$ equal to the identity matrix, since each variable is equivalent to any other. In the realistic OGSTM-BFM application, we adopted a normalization matrix to represent $\mathbf{\Lambda}_i$ (Section 4.2.3) to properly scale the different
biogeochemical variables. However, this strategy implies that the most relevant components are composed by variables that have a high correlation with as many other variables as possible, including processes that can be less relevant for an analysis more interested in the photic layer (e.g., deep layer processes). Alternative options can be investigated and, for instance, similarly to scaling strategies applied in sampling methods proposed in literature (Dovera and Della Rossa (2012)), $\mathbf{\Lambda}_i$ could be designed to focus on the most relevant processes. In particular, $\mathbf{\Lambda}_i$ can be a sparse matrix with non-zero coordinates corre-
sponding to variables and regions of interest based on information derived by a climatology or a multi-annual simulation.

As for $\mathbf{\Lambda}_i$, a preliminary approach is applied for the definition of the model error covariance matrix $\mathbf{Q}_i$ (with its local counterpart $\mathbf{Q}_i^p$) used in the 3D realistic GHOSH implementation (Section 4.2). The diagonal matrix $\mathbf{Q}_i$ is not very representative of the model error covariances of the relatively complex OGSTm-BFM model. A more advanced parameterization of $\mathbf{Q}_i^p$ deserves to be further investigated for future applications (e.g., using a non-diagonal low rank matrix as in 3D-variational
methods, Teruzzi et al. (2014)).

Compared to other ensemble filters, the key feature of the GHOSH filter is the higher polynomial order of its sampling method. The advantage of a higher order is not limited to a better estimation of the mean state but it extends to the covariance matrix of the error probability distribution. In fact, the polynomial order in the estimation of the covariance matrix is half the order of the mean (not shown). Since the covariance matrix is involved in the state estimation, the more accurate GHOSH
results shown in Section 5 can be partly related to an improvement in the error covariance matrix order of approximation.

In addition to an higher polynomial order, the GHOSH filter features a resampling taking place twice per step. Ensemble Kalman-like filters usually adopt an interpolation strategy by applying the observation operator directly to the $\tilde{\mathbf{X}}_i^f$ ensemble of the states after the model evolution (see equation (28)). However, this strategy might lead to inaccurate estimations because the model error covariance matrix $\mathbf{Q}_i$ is not taken into account in the interpolation. For this reason, GHOSH applies the observation
operator to a new ensemble, resampled between the forecast and analysis phases (Fig. 2). In this way, $\mathbf{Q}_i$ is taken into account and, especially in the case of a nonlinear observation operator, the high order of the sampling method is exploited to improve the accuracy of the projection into the observation space.

Similarly to other ensemble filters, a weighted ensemble is used in the GHOSH filter. However, the GHOSH filter differs substantially from the particle filter (Bocquet et al. (2010)) and other types of weighted ensemble filters, since the latters
evolve the value of the weights, whilst GHOSH keeps them fixed and resamples the ensemble to find new members that are well-suited for each (not evolving) weight. Adopting this approach, the problem of some particle weights quickly tending to zero, that usually affects particle filters (van Leeuwen et al. (2019)) can be avoided.





Finally, as in most ensemble methods, the computational cost of the GHOSH filter mainly depends on the ensemble size (Nerger et al. (2005)). The performance of GHOSH and SEIK implemented in Lorenz96 and the profiling of the GHOSH

implemented in OGSTM showed that the cost of the GHOSH is no more a concern than in other order 2 data assimilation schemes (5). The benefits of the higher order offered by the GHOSH filter do not depend on a bigger computational effort, instead they rely on a mathematical advantage. In fact, the GHOSH allows the exploitation of the degrees of freedom in the randomness of the sampling process, choosing ensembles that have a better consistency with the error distribution.

## 7   Conclusions

This work presented a novel ensemble hybrid DA filter (GHOSH). The GHOSH filter features a higher order of approximation compared to other ensemble filters, which resulted in better data assimilation performances.

The order of an ensemble method is one of the most relevant proxies of a filter skill. In fact, the order 2 schemes (e.g., SEIK, ETKF, ESTKF) are peferred nowadays to older order 1 methods (e.g., SEEK). This study represents a natural further step in this direction, opening to a new class of data assimilation schemes with order greater than 2.

The outstanding improvements of the high order GHOSH filter with respect to the SEIK filter are demonstrated in an extensive set of twin experiments based on the Lorenz96 model. Higher RMSE reductions (GHOSH error up to three times smaller than SEIK) occurred in case of small inflation, large number of observations or large ensemble size. Considering each filter tuned with its optimal forgetting factor, the GHOSH filter showed improved capacity to take into account non-linearities by a lesser need of inflation with respect to SEIK, and achieved the best RMSE reductions (GHOSH error lower than half of

the SEIK error) in cases of moderate ensemble size.

The feasibility of the GHOSH filter in a realistic geophysical application has also been assessed by testing a 3D parallel implementation of a Mediterranean physical-biogeochemical model. The GHOSH improved the agreement between the forecast and observations without producing unrealistic effects on the non-assimilated variables. Considering relevant marine ecosystem dynamics, GHOSH filter successfully constrained the uncertainty of the phytoplankton winter bloom events while

assimilating ocean color. Furthermore, a sensitivity analysis indicates that the use of a higher order has positive impacts on the performance of a non-assimilated variable (e.g., nutrients).

From the computational point of view, the implementation of the GHOSH filter proved to be as demanding as the SEIK, given that the computational cost is dominated by the integration of the model and by the ensemble size.

*Code availability.* A GHOSH python implementation is available from the GitHub page: https://github.com/Sword-Code/PythonDA under

the licence GNU GPLv3. The exact version used to produce the twin experiment results and the related plots (Section 5.1) is archived on Zenodo (https://doi.org/10.5281/zenodo.8219559). The FORTRAN implementation of the GHOSH is based on the OGSTM model (available at https://github.com/inogs/ogstm) coupled with the BFM model (available at https://bfm-community.github.io/www.bfm-community.eu/). The exact version of the OGSTM-BFM-GHOSH system used to produce results and plots described in Section 5.2 is archived on Zenodo (https://doi.org/10.5281/zenodo.10019279).

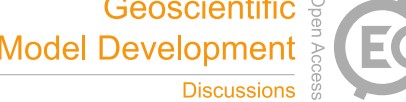



## Appendix A:  High-order sampling

Sampling a limited number of ensemble members that effectively represent the uncertainty of the state estimation is a crucial part of any ensemble algorithm. The GHOSH filter uses multi-dimensional Gauss-Hermite-like quadrature rules to choose the ensemble members, achieving a high polynomial order of convergence.

The core idea behind the GHOSH sampling can be proved by taking two independent random variables with the same moments up to a certain order $h$. Then, they have the same mean after applying any polynomial operator of order $h$. In fact, if $\mathcal{F} : \mathbb{R}^s \longrightarrow \mathbb{R}$ is a polynomial operator such that

$$\mathcal{F}(\boldsymbol{x}) = \sum_{\xi=0}^{h} \sum_{j_1,\ldots,j_\xi=1}^{s} f_\xi^{j_1,\ldots,j_\xi} x_{j_1} \cdots x_{j_\xi}, \tag{A1}$$

where $s$ is the space dimension, $f_\xi^{j_1,\ldots,j_\xi}$ are the polynomial coefficients and $x_1,\ldots,x_s$ are the coordinates of $\boldsymbol{x}$, then the mean $\bar{\mathcal{F}}$,

$$\bar{\mathcal{F}} = \int_{\mathbb{R}^s} \mathcal{F}(\boldsymbol{x}) p(\boldsymbol{x}) dx = \sum_{\xi=0}^{h} \sum_{j_1,\ldots,j_\xi=1}^{s} f_\xi^{j_1,\ldots,j_\xi} \int_{\mathbb{R}^s} x_{j_1} \cdots x_{j_\xi} p(\boldsymbol{x}) d\boldsymbol{x}, \tag{A2}$$

depends only on the first $h$ moments of the probability density function $p$, which are represented by the last integral in equation (A2).

This proves that the mean of an operator can be computed exactly up to a certain order $h$ by substituting the sampled probability distribution with any discrete finite probability distribution (i.e., a weighted ensemble) as long as their moments match (up to order $h$).

In order to build an ensemble with this property, the moment-matching equations are summarized in the nonlinear system

$$\begin{cases} \vdots \\ \sum_{m=1}^{r+1} x_{j_1}^m \cdots x_{j_\xi}^m w_m = \mu_{j_1,\ldots,j_\xi}, \\ \vdots \end{cases} \tag{A3}$$

with one equation for each $\xi \in \{0,\ldots,h\}$ and for each $j_1,\ldots,j_\xi \in \{1,\ldots,s\}$.

In system (A3), $x_j^m$ is the $j$-th coordinate of the $m$-th ensemble member, $r+1$ is the ensemble size, $w_m \geq 0$ are the weights and $\mu_{j_1,\ldots,j_\xi}$ are the statistical moments of $p$, i.e.,

$$\mu_{j_1,\ldots,j_\xi} = \int_{\mathbb{R}^s} x_{j_1} \cdots x_{j_\xi} p(\boldsymbol{x}) d\boldsymbol{x}. \tag{A4}$$

The system is solved considering $x_j^m$ and $w_m$ as the unknowns, while $p$ can be taken uncorrelated, normalized and with $0$ mean without loss of generality. In fact, the ensemble matrix $\tilde{\mathbf{X}} = (x_j^m)$ (which contains in its columns the ensemble members with $0$ mean and covariance matrix equal to the identity matrix $\mathbf{I}_s$) can be transformed in the ensemble matrix $X$ with arbitrary





mean $\bar{x}$ and covariance matrix $LL^T$ through the projection

$$\mathbf{X} = \bar{x}\mathbb{1}_{1\times(r+1)} + \mathbf{L}\tilde{\mathbf{X}}, \tag{A5}$$

where $\mathbb{1}_{1\times(r+1)}$ is a matrix (of the subscripted size) filled with ones.

In the special case of the standard normal distribution, i.e., $p(\boldsymbol{x}) = \mathcal{N}(\boldsymbol{x}, 0, \mathbf{I}_s)$, the moments $\mu_{j_1,\ldots,j_\xi}$ are given by

$$\int_{\mathbb{R}^s} x_1^{k_1}\cdots x_s^{k_s}\mathcal{N}(\boldsymbol{x};0,\mathbf{I}_s)\,d\boldsymbol{x} = 0 \tag{A6}$$

if any exponent $k_j$ is an odd number, or

$$\int_{\mathbb{R}^s} x_1^{k_1}\cdots x_s^{k_s}\mathcal{N}(\boldsymbol{x};0,\mathbf{I}_s)\,dx = \frac{k_1!}{\sqrt{2}^{k_1}\left(\frac{k_1}{2}!\right)}\cdots\frac{k_s!}{\sqrt{2}^{k_s}\left(\frac{k_s}{2}!\right)} \tag{A7}$$

otherwise. Further, the solutions $x_j^m$ and $w_m$ are the nodes and weights of the Gauss-Hermite quadrature rule.

In general, by applying to system (A3) the substitution

$$\begin{cases} w_m = v_m^2, \\ x_j^m = \dfrac{u_j^m}{v_m}, \end{cases} \tag{A8}$$

the weights are naturally forced to be positive, and the system can be conveniently rewritten as

$$\begin{cases} \vdots \\ \displaystyle\sum_{m=1}^{r+1} u_{j_1}^m\cdots u_{j_\xi}^m v_m^{2-\xi} = \mu_{j_1,\ldots,j_\xi}, \\ \vdots \end{cases} \tag{A9}$$

which, in the case of order $h = 2$, can be expressed in matrix form as

$$\left(\begin{array}{c|c} \boldsymbol{v} & \boldsymbol{\Omega}^h \end{array}\right)^T \left(\begin{array}{c|c} \boldsymbol{v} & \boldsymbol{\Omega}^h \end{array}\right) = \mathbf{I}_s, \tag{A10}$$

where $\boldsymbol{v} \in \mathbb{R}^{r+1}$ is the column vector with coordinates $v_m$ and $\boldsymbol{\Omega}^h$ is an $(r+1)\times s$ matrix with coordinates $u_j^m$.

Note that equation (A10) can be solved for any unit vector $\boldsymbol{v}$ and for any value of $s \leq r$. If constant weights are chosen and $s = r$, then this method is equivalent to the second-order exact sampling of the SEIK filter (Pham (1996)), which leads to, in the formalism of equation (A5),

$$\mathbf{X} = \bar{x}\mathbb{1}_{1\times(r+1)} + \mathbf{L}\boldsymbol{\Omega}^{hT}\mathbf{W}^{-\frac{1}{2}}, \tag{A11}$$

where $\mathbf{W} = \mathrm{diag}(\boldsymbol{w})$ is the $(r+1)\times(r+1)$ diagonal matrix of weights.





In general, if $h > 2$, $s$ must be lower than $r$ to obtain a solution to system (A9). The obtained $s$-dimensional ensemble can be extended to an $r$-dimensional one: its projection on the $s$-dimensional subspace defined by the first $s$ coordinates will retain order $h$, while the remaining components are sampled with order 2. This is achieved by starting from $v$ and $\mathbf{\Omega}^h$, given by a particular solution of system (A9). Then, exploiting symmetry and rotational invariance, randomness is added to avoid biases by multiplying $\mathbf{\Omega}^h$ by any random $s \times s$ orthogonal matrix $\mathbf{\Omega}^{rnd}$. Finally, the $(r+1) \times (r-s)$ matrix $\mathbf{\Omega}^\perp$ is chosen to fulfil


$$\left( \begin{array}{c|c|c} v & \mathbf{\Omega}^h \mathbf{\Omega}^{rnd} & \mathbf{\Omega}^\perp \end{array} \right)^T \left( \begin{array}{c|c|c} v & \mathbf{\Omega}^h \mathbf{\Omega}^{rnd} & \mathbf{\Omega}^\perp \end{array} \right) = \mathbf{I}_{r+1}. \tag{A12}$$

The sampling matrix $\mathbf{\Omega}$ is defined as

$$\mathbf{\Omega} = \left( \begin{array}{c|c} \mathbf{\Omega}^h \mathbf{\Omega}^{rnd} & \mathbf{\Omega}^\perp \end{array} \right). \tag{A13}$$

Now the columns of $\mathbf{\Omega}^T W^{-\frac{1}{2}}$ represent an $r$-dimensional ensemble of order $h$ in the first $s$ coordinates and order 2 in the last $r - s$ coordinates. Equation (A11) needs to be modified to properly project the higher order subspace in the most relevant

components of the covariance matrix $\mathbf{LL}^T$. It can be proved that such components, ordered by relevance, are the column of the matrix $\mathbf{LC}^T$, where $\mathbf{C}$ is a $r \times r$ orthogonal change-of-basis matrix such that

$$\mathbf{L}^T \mathbf{\Lambda}^{-1} \mathbf{L} = \mathbf{C}^T \mathbf{E} \mathbf{C}. \tag{A14}$$

In the last equation, the right-hand side is an eigenvalue decomposition of the left-hand side, with $E$ being an $r \times r$ diagonal matrix of eigenvalues in descending order and $\mathbf{\Lambda}$ being an appropriate $N \times N$ matrix. The purpose of $\mathbf{\Lambda}$ has already been

discussed in Section 2.2.

After the change of basis induced by $C$, the most relevant components are on the left side of the matrix product $LC^T$, which is compliant with the ensemble stored in $\mathbf{\Omega}^T W^{-\frac{1}{2}}$, and the sampling is finally obtained by

$$\mathbf{X} = \bar{\boldsymbol{x}} \mathbb{1}_{1 \times (r+1)} + \mathbf{LC}^T \mathbf{\Omega}^{h^T} \mathbf{W}^{-\frac{1}{2}}. \tag{A15}$$

*Author contributions.* GC, S Salon and S Spada designed the study. S Spada developed the algorithms, supervised by SM. S Spada, AT and
GC designed the experiments. S Spada implemented the code and performed the simulations collaborating with AT for the 3D case. S Spada wrote the manuscript, with the contribution of AT, GC and CS. All the authors participated in acquiring the funding for the project.

*Competing interests.* The authors declare no competing interests.



*Acknowledgements.* We acknowledge the CINECA award under the ISCRA initiative and the HPC-TRES program, for the availability of high-performance computing resources.

This study has been conducted using E.U. Copernicus Marine Service data.

The research reported in this work was supported by OGS and by the SEAMLESS project (https://www.seamlessproject.org/).

This work was partly funded by the European Union's Horizon 2020 research and innovation programme under grant agreement No 101004032 (project SEAMLESS).




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
