# Peer review of "GHOSH v1.0.0: a novel Gauss-Hermite High-Order Sampling Hybrid filter for computationally efficient data assimilation in geosciences"

_Geoscientific Model Development, 2023_

## Referee Comment (RC2)

**Reviewer report for gmd-2023-170 "GHOSH v1.0.0: a novel Gauss-Hermite High-Order Sampling Hybrid filter for computationally efficient data assimilation in geosciences"**

This work proposes an ensemble filter (GHOSH) that conducts sampling in such a way that the resulting sample statistics can match the moments of a target distribution up to a specified order. The moment-matching trick is based on Hermite polynomial approximations to the underlying functions, whereas the target distribution is set to be Gaussian. The derived filter is tested in two examples, both indicating that the GHOSH outperforms an existing filter (SEIK) for the experiments conducted in the current work.

The manuscript is clearly written and reasonably organized in general. Below is a list of minor-to-moderate issues spotted in the current manuscript.

**Spotted issues**

1. Page 1 –

   - Line 2: Consider replacing "one of" by "among" or something similar, since "algorithms" is the subject.
   - Line 20: What does "a higher order of convergence" mean here?

2. Page 2 –

   - Line 42: "Montecarlo" → "Monte Carlo".
   - Line 51 – 52: Rephrase the sentence "the second order approximation is more effective the closer the ensemble members are to each other, thus, the larger the ensemble spread the worse will be the approximation error in the mean computation."

3. Line 58 – 59, Page 3: The "$2r + 1$ ensemble members" requirement does not appear exact. For the unscented transform, one can use either principal

component analysis (PCA) or truncated singular value decomposition (TSVD) to reduce the number of ensemble members, see

`https://doi.org/10.1175/2008JAS2681.1`
`https://doi.org/10.1016/j.physd.2008.12.003`

It may be worth discussing the similarities and differences between the ideas used to control the number of ensemble members in the aforementioned works and the current manuscript.

4. Line 109 – 117, Page 5: The discussion on the extension of GHOSH to more generic distributions makes sense. A missing part, however, is that the authors did not explain why they confine themselves to Gaussian distribution in the current work, and what could be the challenges for the generalization of GHOSH to more generic distributions

5. Eq. 38, Page 13: The notation w.r.t the $Q_i$ component is somewhat confusing. I guess it should be $(Q_i^p)^{-1}$, but it looks like $Q_i^{p-1}$.

6. In the experiments w.r.t the Lorenz96 model, localization does seem used. What is the reason behind this setting?

7. Line 434 – 435, Page 19: Why "it implies that the PCA measures the Pearson correlation"?

8. Line 490, Page 21: If I've understood correctly, the "best" label corresponds to the configuration that leads to the best DA performance. If so, then in Figure 4 one should use one block to represent it, and I don't see the point to use a single row for the representation.

9. Line 661, Page 33: "an higher" $\rightarrow$ "a higher".

---

## Author Comment (AC2)

**Review of "GHOSH v1.0.0: a novel Gauss-Hermite High-Order Sampling Hybrid filter for computationally efficient data assimilation in geosciences"**

**Reply on Reviewer comment RC1**

Summary: The paper present an interesting idea of using high-order sampling to improve the accuracy of higher statistical moments of the ensemble, therefore achieving better skill in ensemble prediction. I was expecting to learn some details about the new approach, but to my disappointment I am still quite confused after reading the paper. In my opinion, the paper tries to achieve too many things at once: introduce the method mathematically and show experiment results in two different models. No wonder that you don't have enough room to provide the details. In the methodology, many steps in the formula are stated as "can be proven" but with no guidance for readers to actually understand the logic (skipping too many steps in a proof). Ideally, when introducing a new method, I would expect a paper focused on the theory and illustrations using simple models; then followed by other paper(s) utilizing the method in bigger models describing how you solved any additional issues in implementation. The current paper did the introduction poorly, I suggest the authors to give it a thorough revision, at least address the following major issues.

**As previously done in the short reply, we thank the Reviewer for the comments and suggestions, which highlight the need to state more clearly the goal and novelty of our manuscript, and of improving its readability. Hereafter, our point-by-point responses to the Reviewer's comments are provided in bold green.**

**We will surely keep this in consideration in the revised version of the manuscript. Firstly, we would like to clarify the main objective of the work, since it appears this was not clear enough. We are introducing a new method (GHOSH filter) that is focused on the use of a higher order of approximation to evaluate the ensemble mean in data assimilation applications. We are not evaluating standard deviation, skewness, kurtosis and higher statistical moments of the ensemble in the present work.**

**On the other hand, it is true that the equality of the statistical moments up to order h implies an approximation of the mean up to order h, as demonstrated in Appendix A and discussed later in the reply to the Reviewer's comment 2.**

**Secondly, as highlighted also by the Reviewer #3, we recognize that we have a lot of material, and possibly too much for a single paper. We received a positive answer by the Topic Editor about a request we sent in the past weeks on the possibility to split the manuscript in two parts, and we will prepare a revised version of the manuscript divided in two parts.**

Major Issues:

1. It's unclear what is the actual size of the state vector now you are employing a weighted ensemble approach. When you state that you use 2r members to achieve 3rd-order sampling, such as the Wan & van der Merwe 2000 paper, the members are paired with mean +- r perturbations, does it mean you only need to store r+1 numbers?

**If the state vector has dimension N, then an ensemble method with ensemble size m must store m state vectors, for a total of m times N numbers. If the ensemble is weighted, also the weights must be considered, so the total becomes mN + m, as in the particle filter (PF) case. Differently from PF, the GHOSH weights are constant in time, thus the degrees of freedom are mN. In Wan & van der Merwe 2000, they use an ensemble size of m=2r+1 in the attempt of achieving a 3rd-order samping. In comparison, ETKF and SEIK use m=r+1 ensemble members for their 2nd-order sampling. The GHOSH novel strategy tries to exploit the best of the two approaches: it uses only m=r+1 members (as the 2nd-order methods), but those members are chosen in such a way that in the principal components the order of the sampling is higher than 2. The simplest example of the last statement in the case of a 2-dimensional standard normal error is the following 2nd-order sampling, achieving 3rd-order in the x direction:**

$P1: (x1, y1) = (0, \sqrt{2}),$

$P2: (x2, y2) = (\sqrt{3/2}, - \sqrt{2}/2),$

$P3: (x3, y3) = (- \sqrt{3/2}, - \sqrt{2}/2).$

**In fact:**

**the 1st moment associated to x (i.e., the mean along x) is:**
$(x1 + x2 + x3)/3 = (0 + \sqrt{3/2} + (- \sqrt{3/2}))/3 = 0,$

**the 1st moment associated to y (i.e., the mean along y) is:**
$(y1 + y2 + y3)/3 = (\sqrt{2} + (- \sqrt{2}/2) + (- \sqrt{2}/2))/3 = 0,$

**the 2nd moment associated to x^2 (i.e., the variance of x) is:**
$(x1^2 + x2^2 + x3^2)/3 = (0 + 3/2 + 3/2)/3 = 1,$

**the 2nd moment associated to y^2 (i.e., the variance of y) is:**
$(y1^2 + y2^2 + y3^2)/3 = (2 + 1/2 + 1/2)/3 = 1,$

**the 2nd moment associated to xy (i.e., the covariance between x and y) is:**
$(x1 \cdot y1 + x2 \cdot y2 + x3 \cdot y3)/3 = (0 \cdot \sqrt{2} + \sqrt{3/2} \cdot (- \sqrt{2}/2) + (- \sqrt{3/2}) \cdot (- \sqrt{2}/2))/3 = 0$,

**the 3rd moment associated to x^3 is:**
$(x1^3 + x2^3 + x3^3)/3 = (0 + \sqrt{3/2}^3 + (- \sqrt{3/2})^3)/3 = 0.$

**Thus, the ensemble P1, P2, P3 is compliant with the statistical moments of a standard normal distribution up to order 2, and also have the same 0-skewness along the x-axis.**

**In the manuscript, we will include a new Appendix C to guide the reader through some simple examples (as the one before) in order to improve the manuscript readability also for people without a strong mathematical background (see also answers to the Reviewer's next comment).**

The weights are computed according to A3 which is a nonlinear system itself. What is the computational complexity of this step? You claimed that the GHOSH filter is no more costly than the 2nd-order EnKFs with same ensemble size, but can you provide some complexity arguments for each step in your filter? For example, given ensemble size m, obs count p and model state size n, the ETKF has complexity $m^2 p + m^3 + m^3 n$, according to Tippett 2003 (doi: 10.1175/1520-0493(2003)131<1485:ESRF>2.0.CO;2). Although with fixed ensemble size, the filter is doing additional steps (e.g. the weight computation). I'm not convinced that they are comparable.

**The fact that GHOSH is no more costly than the 2nd-order EnKFs takes into account that the nonlinear system where weights are computed needs to be solved only once. Its computational cost depends on the chosen nonlinear solver. It should also be noted that, when available, analytical solutions should be preferred. During the initialization of the GHOSH filter, the provided python code uses an analytical constructive method to solve the system up to order 5 in $O(m^2)$ operations.**

**The GHOSH computational complexity is the same as ETKF, i.e., $O(m^2 p + m^3 + m^2 n)$, since the GHOSH filter requires a few more operations per step (e.g., resampling and eigenvalue decomposition) that are $O(m^3 + m^2 n)$ and do not add to the computational complexity. This means that, even if the GHOSH filter needs a little more computational time, it scales as the ETKF, the SEIK, or similar EnKF. Further, in realistic applications, the EnKF computational cost is usually widely dominated by the model integration cost (and, secondly, by I/O), while forecast and analysis do not represent a concern at all.**

**In the manuscript, we will follow the Reviewer suggestion of evaluating explicitly the computational complexity. In addition, we will also discuss timings for the two filters in the Lorenz twin experiment (according also to the Reviewer#3 comments). In particular, we will modify L. 514 as follows:**

**"From the computational point of view, the whole experiment set needed around $9$ computational hours. SEIK and GHOSH schemes used 12% and 16% of the total time respectively, while the rest was committed to model integration. The GHOSH filter executes more operations than SEIK (e.g., an eigenvalue decomposition) resulting in more computational time even if the asymptotic computational complexity of the two methods is the same. However, even in the case of a relatively simple model like the Lorenz96, the time to solution is dominated by the model integration and the difference between GHOSH and SEIK only accounts for 4% of the total time. Unexpectedly, the model integration time (averaged every 100 twin experiments) when**

**applying the SEIK filter sometimes is longer than the GHOSH filter case, up to twice the time, depending on some settings and random factors."**

2. Fig. 1 provides some visual guide to understand the statement about "m sample points are required to represent a hth-order approximation of an error distribution in r-dimensional space". I can see that in 3D you need 4 points to span a volume (the covariance), and I can see that you need 4 degrees of freedom to provide a unique 3x3 covariance matrix. But from this point onward, I don't understand anymore.

First, the error distribution depicted is symmetric, meaning there is only the mean and identity covariance, there is not even correlation between variables in different dimensions, how can I visualize the higher-order moments (skewness)?Therefore, I don't see how the 4 points are giving accurate 3rd moment in 2D, there is no skewness anyway in the picture.

**Considering this and the previous Reviewer's comments and comments from Reviewer #3, we propose to describe Fig. 1 in more detail in a novel appendix (Appendix C) adding the description (including equations and calculations) of the concepts summarised in Fig. 1 panels (e.g., computation of the moments). Indeed, this part of the manuscript is too concise and a reader could really benefit from additional details.**

**Appendix C will show all the detailed computations needed to ensure that a reader could follow each step. For example, talking of 4 points (P1, P2, P3 and P4) giving accurate 3rd moments in 2D, the reader will be guided as follows:**

$P1: (x1, y1) = (\sqrt{2}, 0),$

$P2: (x2, y2) = (0, \sqrt{2}),$

$P3: (x3, y3) = (-\sqrt{2}, 0),$

$P4: (x4, y4) = (0, -\sqrt{2}).$

**The 1st moment associated to x (i.e., the mean along x) is:**
$(x1 + x2 + x3 + x4)/4 = (\sqrt{2} + 0 + (-\sqrt{2}) + 0)/4 = 0,$

**the 1st moment associated to y (i.e., the mean along y) is:**
$(y1 + y2 + y3 + y4)/4 = (0 + \sqrt{2} + 0 + (-\sqrt{2}))/4 = 0,$

**the 2nd moment associated to x^2 (i.e., the variance of x) is:**
$(x1^2 + ... + x4^2)/4 = (2 + 0 + 2 + 0)/4 = 1,$

**the 2nd moment associated to y^2 (i.e., the variance of y) is:**
$(y1^2 + ... + y4^2)/4 = (0 + 2 + 0 + 2)/4 = 1,$

**the 2nd moment associated to xy (i.e., the covariance between x and y) is:**

$(x1 \cdot y1 + ... + x4 \cdot y4)/4 = (\sqrt{2} \cdot 0 + 0 \cdot \sqrt{2} + \sqrt{2} \cdot 0 + 0 \cdot \sqrt{2})/4 = 0,$

**the 3rd moment associated to x^3 is:**

$(x1^3 + ... + x4^3)/4 = (\sqrt{2}^3 + 0 + (-\sqrt{2})^3 + 0)/4 = 0,$

**the 3rd moment associated to x^2 y is:**

$(x1^2 \cdot y1 + ... + x4^2 \cdot y4)/4 = (2 \cdot 0 + 0 \cdot \sqrt{2} + 2 \cdot 0 + 0 \cdot 2)/4 = 0,$

**the 3rd moment associated to x y^2 is:**

$(x1 \cdot y1^2 + ... + x4 \cdot y4^2)/4 = (\sqrt{2} \cdot 0 + 0 \cdot 2 + \sqrt{2} \cdot 0 + 0 \cdot 2)/4 = 0,$

**the 3rd moment associated to y^3 is:**

$(y1^3 + ... + y4^3)/4 = (0 + \sqrt{2}^3 + 0 + (-\sqrt{2})^3)/4 = 0.$

**Thus, the ensemble P1, P2, P3, P4 is compliant with the statistical moments of a standard normal distribution up to order 3.**

Second, why does projection from high-dimensional to low-dimensional space relate to a higher-order sampling in the low-dimensional space? Somehow these 2 members project to the same member in lower dimension, which means they are paired in a way that projection will cancel out the perturbations and only leave the mean? But why is that equivalent to a weighted member, what is the weight in this 3-point case anyway? Linking back to the 4 points in 3D, are the positioning only allowed to be squares with no rotation? In the 3D case the tetrahedron is allowed to rotate arbitrarily. Here in 2D if the square is rotated they project to 4 points in 1D.

**We totally agree with the Reviewer, each tetrahedron rotation implies a different shadow (i.e., projection). Only a few possible rotations of the tetrahedron project a perfect square in the xy plane, and even fewer rotations project a square that further projects into 3 points (instead of 4) in the x axis. The strength of GHOSH is exactly this: being able to catch the right rotation to achieve the best possible projections. Or, equivalently, the GHOSH sampling chooses ensemble members matching high order moments of the error distribution along some chosen directions (i.e., the principal components of the error), obtaining a better approximation of the mean along those directions.**

**Concerning the other Reviewer's comments about projections, please read the next response.**

Finally, why does three weighted samples in 1D give a 5th order approximation? I don't see any moments in the sample distribution higher than the 2nd.

At this point, I would rather you don't show this figure at all, but replace Lines 89-108 with a more step-by-step reasoning, or formal proof, of the statement such as "2r samples give a 3rd-order approximation of r-dimensional space".

As proposed above, the addition of the Appendix C will provide a step-by-step reasoning that will help the reader to understand the issues raised by the Reviewer. The detailed description of Appendix C will provide an explanation of what is shown in Fig. 1, including a practical example of a 5th order weighted ensemble in 1D composed by 3 points P1, P2 and P3, i.e.:

$P1: x1 = \sqrt{3};\ w1 = 1/6,$

$P2: x2 =- \sqrt{3};\ w2 = 1/6,$

$P3: x3 = 0;\ w3 = 2/3.$

The first moment (mean) is:

$w1 \cdot x1 + w2 \cdot x2 + w3 \cdot x3 = 1/6 \cdot \sqrt{3} + 1/6 \cdot (- \sqrt{3}) + 2/3 \cdot 0 = 0,$

the second moment (variance) is:

$w1 \cdot x1^2 + w2 \cdot x2^2 + w3 \cdot x3^2 = 1/6 \cdot 3 + 1/6 \cdot 3 + 2/3 \cdot 0 = 1,$

the third moment (skewness) is:

$w1 \cdot x1^3 + w2 \cdot x2^3 + w3 \cdot x3^3 = 1/6 \cdot \sqrt{3}^3 + 1/6 \cdot (- \sqrt{3})^3 + 2/3 \cdot 0 = 0,$

the fourth moment (kurtosis) is:

$w1 \cdot x1^4 + w2 \cdot x2^4 + w3 \cdot x3^4 = 1/6 \cdot 9 + 1/6 \cdot 9 + 2/3 \cdot 0 = 3,$

the fifth moment is:

$w1 \cdot x1^5 + w2 \cdot x2^5 + w3 \cdot x3^5 = 1/6 \cdot \sqrt{3}^5 + 1/6 \cdot (- \sqrt{3})^5 + 2/3 \cdot 0 = 0.$

Thus, the ensemble P1, P2, P3 is compliant with the statistical moments of a standard normal distribution up to order 5.

The property of this 1-dimensional 5th-order sampling, can be extended to a 2-dimensional ensemble, aimed to keep order 5 along the x-axis while achieving order 3 in the xy plane. Such 4-members ensemble is:

$P1: (x1, y1) = (\sqrt{3}, 0);\ w1 = 1/6,$

$P2: (x2, y2) = (- \sqrt{3}, 0);\ w2 = 1/6,$

$P3: (x3, y3) = (0, \sqrt{3/2});\ w3 = 1/3,$

$P3: (x4, y4) = (0, - \sqrt{3/2});\ w4 = 1/3.$

Note that all the moments along the x-axis are already computed above up to order 5, since this ensemble and the previous one are indistinguishable looking only at the x-coordinate. In fact, considering the projection on the x-axis, P3 and P4 behave as a unique member with weight w3+w4 because they have the same x-coordinate x3=x4. It remains to be checked if the moments involving y match the moments of standard normal distribution up to order 3, i.e.:

the 1st moment associated to y (i.e., the mean along y) is:

$$w_1 \cdot y_1 + w_2 \cdot y_2 + w_3 \cdot y_3 + w_4 \cdot y_4 =$$

$$= 1/6 \cdot 0 + 1/6 \cdot 0 + 1/3 \cdot \sqrt{3/2} + 1/3 \cdot \left(- \sqrt{3/2}\right) = 0,$$

the 2nd moment associated to y^2 (i.e., the variance of y) is:

$$w_1 \cdot y_1^2 + \ldots + w_4 \cdot y_4^2 = 1/6 \cdot 0 + 1/6 \cdot 0 + 1/3 \cdot 3/2 + 1/3 \cdot 3/2 = 1,$$

the 2nd moment associated to xy (i.e., the covariance between x and y) is:

$$w_1 \cdot x_1 \cdot y_1 + \ldots + w_4 \cdot x_4 \cdot y_4 =$$

$$= 1/6 \cdot \sqrt{3} \cdot 0 + 1/6 \cdot \left(- \sqrt{3}\right) \cdot 0 + 1/3 \cdot 0 \cdot \sqrt{3/2} + 1/3 \cdot 0 \cdot \left(- \sqrt{3/2}\right) = 0,$$

the 3rd moment associated to x^2 y is:

$$w_1 \cdot x_1^2 \cdot y_1 + \ldots + w_4 \cdot x_4^2 \cdot y_4 =$$

$$= 1/6 \cdot 3 \cdot 0 + 1/6 \cdot 3 \cdot 0 + 1/3 \cdot 0 \cdot \sqrt{3/2} + 1/3 \cdot 0 \cdot \left(- \sqrt{3/2}\right) = 0,$$

the 3rd moment associated to x y^2 is:

$$w_1 \cdot x_1 \cdot y_1^2 + \ldots + w_4 \cdot x_4 \cdot y_4^2 =$$

$$= 1/6 \cdot \sqrt{3} \cdot 0 + 1/6 \cdot \left(- \sqrt{3}\right) \cdot 0 + 1/3 \cdot 0 \cdot 3/2 + 1/3 \cdot 0 \cdot 3/2 = 0,$$

the 3rd moment associated to y^3 is:

$$w_1 \cdot y_1^3 + \ldots + w_4 \cdot y_4^3 = 1/6 \cdot 0 + 1/6 \cdot 0 + 1/3 \cdot \sqrt{3/2}^3 + 1/3 \cdot \left(- \sqrt{3/2}\right)^3 = 0.$$

**The dimensions can be increased once more by building in a similar way a 3-dimensional ensemble, as shown in Fig. 1. This ensemble will be indistinguishable from the previous one in the xy plane projection and capable of achieving 2nd order on the xyz space. Such ensemble is:**

$$P1: (x_1, y_1, z_1) = \left(\sqrt{3}, 0, - \sqrt{2}\right); \ w_1 = 1/6,$$

$$P2: (x_2, y_2, z_2) = \left(- \sqrt{3}, 0, - \sqrt{2}\right); \ w_2 = 1/6,$$

$$P3: (x_3, y_3, z_3) = \left(0, \sqrt{3/2}, \sqrt{2}/2\right); \ w_3 = 1/3,$$

$$P3: (x_4, y_4, z_4) = \left(0, - \sqrt{3/2}, \sqrt{2}/2\right); \ w_4 = 1/3.$$

**All the moments involving x and y are already checked in the previous calculation up to order 5 along x and up to order 3 in the xy plane. It remains to be checked if the moments involving z match the moments of standard normal distribution up to order 2, i.e.:**

**the 1st moment associated to z (i.e., the mean along z) is:**

$w1 \cdot z1 + w2 \cdot z2 + w3 \cdot z3 + w4 \cdot z4 =$

$= 1/6 \cdot \left(-\sqrt{2}\right) + 1/6 \cdot \left(-\sqrt{2}\right) + 1/3 \cdot \sqrt{2}/2 + 1/3 \cdot \sqrt{2}/2 = 0,$

**the 2nd moment associated to z^2 (i.e., the variance of z) is:**

$w1 \cdot z1^2 + \ldots + w4 \cdot z4^2 = 1/6 \cdot 2 + 1/6 \cdot 2 + 1/3 \cdot 1/2 + 1/3 \cdot 1/2 = 1,$

**the 2nd moment associated to xz (i.e., the covariance between x and z) is:**

$w1 \cdot x1 \cdot z1 + \ldots + w4 \cdot x4 \cdot z4 =$

$= 1/6 \cdot \sqrt{3} \cdot \left(-\sqrt{2}\right) + 1/6 \cdot \left(-\sqrt{3}\right) \cdot \left(-\sqrt{2}\right) + 1/3 \cdot 0 \cdot \sqrt{2}/2 + 1/3 \cdot 0 \cdot \sqrt{2}/2 = 0,$

**the 2nd moment associated to yz (i.e., the covariance between y and z) is:**

$w1 \cdot y1 \cdot z1 + \ldots + w4 \cdot y4 \cdot z4 =$

$= 1/6 \cdot 0 \cdot \left(-\sqrt{2}\right) + 1/6 \cdot 0 \cdot \left(-\sqrt{2}\right) + 1/3 \cdot \sqrt{3/2} \cdot \sqrt{2}/2 + 1/3 \cdot \left(-\sqrt{3/2}\right) \cdot \sqrt{2}/2 = 0.$

**All the above proves that the weighted ensemble P1, P2, P3, P4 is a second-order ensemble achieving order 3 in the subspace of the x and y directions and order 5 along the x direction.**

**We are confident that these step-by-step calculations will help the Reader in following the manuscript arguments.**

3. In your Lorenz96 experiments (Fig. 3), based on the ensemble spread compared to the error, the SEIK filter has basically diverged at around t=5. The climatological error is about 4 for the Lorenz96 system and you can see that after t=5 the SEIK filter solution is as bad as random draw from climatology. I would argue that you need to have a more stable setting for both SEIK and GHOSH here to establish a comparison, especially when we know that SEIK should work for the Lorenz96 system. For some cases when the SEIK is stable, can you still show that the new GHOSH filter will achieve higher accuracy or at least perform as well as the SEIK?

**We thank the Reviewer for pointing out that the example in Fig. 3 is poorly chosen, since it shows a situation where SEIK does not improve with respect to climatology. Thanks to that, Figure 3 will be changed in order to represent a better example, as the one represented in Fig. R1. However, single tests can have high variability, and a more reliable and statistically accurate view can be offered by a RMSE computed in a larger set of experiments (as in Fig. R2). Since SEIK and GHOSH present different behaviours depending on other parameters (e.g., ensemble size, observation frequency, forgetting factor), Fig. 4 resumes the whole set of 56000 Lorenz-96 experiments that have been carried out by averaging blocks of 400 experiments with similar setup (but different random parameters, e.g. observations). Moreover, Fig. 4 shows that SEIK was tested also in stable and convergent conditions, i.e., green squares in the two top panels in Fig. 4. In addition, it is worth noting that the comparison between SEIK and GHOSH has also been tested considering the best**

**tuning in terms of inflation for both filters ("best" line in each panel of Fig. 4) always resulting in GHOSH working as good as SEIK or better.**

[Figure]

**Fig. R1. RMSE of SEIK and GHOSH by time based on all variables, results from one twin experiment example: time between observation is 0.15, forgetting factor is 0.6, ensemble size is 31.**

[Figure]

**Fig. R2. RMSE of SEIK and GHOSH by time based on all variables of 100 twin experiments with the following settings: time between observation is 0.15, forgetting factor is 0.6, ensemble size is 31.**

4. In the verification of filter performance for both models, I mainly see the use of RMSE and ensemble spread as error metrics. This means you are still evaluating the first two moments of error distributions. However, the GHOSH filter is motivated by its capability of capturing higher moments in error. Why didn't you show any proof that it is behaving as expected? For example, in the Lorenz96 case you can compute the skewness and kurtosis of the error distribution and compare the two filters. This will truly be a proof of concept, rather than implying the skill from the forecast error in ensemble mean (1st moment).

**As discussed above, the main goal of the GHOSH filter is to achieve higher approximation order on the ensemble mean without focus on higher order moments. Thus, we think that RMSE is a sound metric to compare the GHOSH performances with respect to SEIK.**

Minor Issues:

The use of nested parentheses in citation is a bit confusing. Try use (author year; author year)? Or is there a prefered format for GMD?

**We thank the Reviewer for the suggestion. The GMD format does not require nested parentheses, and we will update all the citations accordingly.**

Line 42: "Monte Carlo"?

**Thanks, we will correct the typo.**

Line 51: Is there a reference for this? Why is closer together samples more suitable for accuracy using second-order approximation?

**The sentence is related to the fact that the nonlinear effects are smaller if the perturbation of the ensemble members is small and hence the second-order approximation holds better. In other words, if the true uncertainty is small, then a second order approximation is affected by a smaller error. According also to Reviewer #2 and #3 comments, we propose to modify the sentence focusing on uncertainty instead of ensemble spread:**

**"Furthermore, the second order approximation is more effective the closer the ensemble members are to each other (i.e., small uncertainty), thus the higher the**

R1.10

**uncertainty the worse will be the approximation error in the mean computation. Since the state estimation is often affected by a relatively high uncertainty in data assimilation geoscience applications, this approximation error may be not negligible.**

Line 54: It's a bit not clear what you mean by "higher order of approximation" here. I think you refer to the higher moments of the distribution that the samples will exactly reproduce after model integration. If I'm not completely missing the point, this can be stated more clearly to help the readers follow.

**Thanks to this Reviewer comment and to issues raised by Reviewer #3,, we will state more clearly the definition of order of approximation:**

**L. 43: "The number of ensemble members can be reduced by adopting deterministic sampling methods (as opposed to stochastic EnKF methods, see e.g. Carrassi et al., 2018). Examples of deterministic EnKF using second-order sampling methods are SEIK (Pham (2001)) and ETKF (Bishop et al. (2001)). Extending the definition of second-order exact methods in Pham (2001), with the term "order" we refer to the polynomial order of approximation of the filter or of its sampling method. Namely, in case of hth-order, the ensemble has the property of providing a forecast mean with no error as long as the evolution function used for forecasting (i.e., the model) is a polynomial of order h. As proven in Appendix A, this is equivalent to sampling an ensemble that preserves (before applying the model) the first h statistical moments."**

Line 57: I can understand why you need r+1 samples to give mean and covariance (2nd order approximation), but it's a bit hard to see why 2r for 3rd order approximation. You say it can be proven, could you give a reference, or include a proof in appendix?

**The details that will be provided in Appendix C will clarify the need of 2r samples for 3rd order approximation.**

Line 59: apart from the unscented Kalman filter, other methods that preserve the 3rd moment of posterior pdf, such as the one by Hodyss 2011 (DOI: 10.1175/2011MWR3558.1), is also worth mentioning here.

**Since the main aim of the GHOSH filter is a higher order of approximation of the mean and not to provide higher-than-second moments, we think that the citation is not useful in the sentence at L. 59. However, it is perfectly suited for discussing the applicability of the GHOSH sampling to non-Gaussian priors and it will be added to Section 6 of the manuscript in a new dedicated paragraph. We kindly thank the Reviewer for the suggested high-quality reference.**

Line 95: Pham 1996 is the introduction paper for SEIK, are you sure this is the right citation for the second-order exact sampling?

**Indeed, the second order exact sampling is the sampling method used in the SEIK filter, as explained in the Appendix of the cited paper. However, as noted also by Reviewer 3, this citation can be improved to a more recent one (Pham, 2001).**

Line 105-108: this paragraph is very vague, we achieve a fifth order approximation when the variance is "bigger" on x than y, can you be more precise exactly when do we achieve fifth order?

**We thank the Reviewer for this comment, that will make the manuscript clearer. The sampling is of order 5 in x, and if the variance on the x coordinate is bigger than on the other coordinates, then the best order of approximation is achieved along the direction that is affected by the largest error (as explained in comment to line 51). The sentence will be modified to be more clear:**

**"This means that, if the z component of the distribution was less relevant than the x and y components (e.g., a non-standard normal distribution with smaller variance on the z coordinate), then our third order approximation mitigates the approximation error in the x and y dimensions where the spread is higher. And, if the variance is bigger on x than y, then the achieved fifth order approximation reduces the error more efficiently where the spread is maximum (x coordinate)"**

Line 112: h-order, shouldn't it be hth-order?

**Thank you for the suggestion. We will use hth-order.**

Line 128: putting index as superscripts are very confusing, like for this equation, is v_m ^{2-xi} raised to 2-xi power or just indexed by 2-xi? Either put a parenthesis around them if they are indices, or just put the index as subscripts. It's also confusing to see a tuple j1,...,j_xi ending up being a single index from 1 to s. Please provide more contex in (1) about other equations, I guess the other equations in the set are matching a different moment indexed by j1,...,j_xi? How many equations are there in total?

**According to the Reviewer comment, all the indexes will be subscripted. Moreover, according to comments provided also by Reviewer 3, the whole system (1) and the following paragraph will be improved. More equations in (1) will be explicitly written to help the manuscript readability and the text will guide the reader to understand equation (1).**

Line 137: is there a formal relationship you can write down for r and s given h? First, you will need r>s to establish the principle components, otherwise there is rank deficiency. But what to expect given h? Previously you stated that for h=2 you need r>=s+1, and for h=3 you

need r>=2s, in the 1D case you showed for h=5, you have r>=4s. I'm wildly guessing here that "r>=(h-1)*s" might be what you imply?

**Unfortunately, the general relation between r, h, and s is non-linear and we are not aware of any closed form to state it. On the other hand, it is true that certain values for r, h and s are valid if and only if system (1) admits a solution. The above relations for h=2 and h=3 holds (changing r with r+1, since probably the Reviewer was referring to the ensemble size). In the case h=5 we know that system (1) has a solution at least for r>=2^(s+1) - 2. This comes from the analytical constructive solution to system (1) implemented in the initialization of the GHOSH filter in the provided python code.**

Line 179: why is the error covariance approximated not exactly given by the ensemble perturbation matrix?

**We thank the Reviewer for pointing out the ambiguity of this statement. The mean and covariance of the analysis is an approximation, since they are derived by the forecast mean and covariance, which are approximations affected by a sampling error. On the other hand, analysis mean and covariance after sampling are exact in the sense that the ensemble is chosen to match those quantities. In order to remove any misunderstanding, we will change the superscripted "a" with "f" in equations (9)-(12).**

Line 247: (26) is the same as (6), you can just remove (26) and say that C and Omega are computed using x^a in (4)-(7). By the way, what is the computational cost for this step? You have a eigenvalue decomposition here so at least O(r^3)?

**Since the focus of the analysis is X^a, which is computed in (26), we prefer to keep that equation, even if it could be a little redundant. Also considering Reviewer #3 comments, line 248 will be modified to make clear that (6) is not the only relevant equation (also equations (4)-(7) play a relevant role). The computational cost of the eigenvalue decomposition depends on the adopted algorithm but it is safe to consider it O(r^3). The resampling step has a computational complexity of O(r^3 + N r^2), where the last term comes from the matrix multiplication with L. The addition of detailed consideration about the computational complexity will help the reader to have an evaluation of the GHOSH costs (see major issue 1).**

Line 296: what is l'?

**In the manuscript we called N the state vector dimension, and n the dimension of the observation vector. The localization process uses three operators. The first one reduces the dimension of the state vector from N to l. The second one reduces the dimension of the observation vector from n to l'. In general, l' is not equal to l, as N is not equal to n. The third one is the delocalization operator, used to build back a N-dimensional state vector.**

Table 1 can be removed, your experiment name already encode the value of the parameters as listed.

**We prefer to keep Table 1, since some experiment names only appear in the table. Moreover, Table 1 is useful for the reader to visualise all the configurations of the 3D experiments.**

Line 365: have you defined the "forgetting factor" anywhere yet? Is it the same one from Pham 1996?

**Yes, the forgetting factor is the same as Pham 1996, and it is defined at line 264. To help the reader, we propose to change line 365 adding a reference to the relevant equation: "Both filters have the same forgetting factor (equation (32))"**

Line 469: what range of values have you tested and how many different experiments have you run for the tuning process beyond the 18 you showed. This will provide an idea of the cost for tuning as well.

**As observed by the Reviewer, the tuning was not limited to the 18 experiments listed in Table 1. However, some of the tuning simulations were very short or have been focused on validating GHOSH effects on a subset of the variables in the biogeochemical model. Moreover, tuning cost can depend on the typical characteristics of the realistic application. Thus, we think that adding the number of runs carried out for tuning the filter will not be very informative for the reader.**

Line 479: Would it help to show the results for Lorenz96 immediately after the description of its experiment design? You can have section 4 focused on the Lorenz results and section 5 on the BGC model results.

**This issue will be solved by splitting the manuscript in two parts.**

**References**

**Pham, D. T. 2001. Stochastic methods for sequential data assimilation in strongly non-linear systems. Mon. Wea. Rev. 129, 1194–1207.**

---

## Author Comment (AC3)

**Reviewer report for gmd-2023-170 "GHOSH v1.0.0: a novel Gauss-Hermite High-Order Sampling Hybrid filter for computationally efficient data assimilation in geosciences"**

**Reply on Reviewer comment RC2**

This work proposes an ensemble filter (GHOSH) that conducts sampling in such a way that the resulting sample statistics can match the moments of a target distribution up to a specified order. The moment-matching trick is based on Hermite polynomial approximations to the underlying functions, whereas the target distribution is set to be Gaussian. The derived filter is tested in two examples, both indicating that the GHOSH outperforms an existing filter (SEIK) for the experiments conducted in the current work.

The manuscript is clearly written and reasonably organized in general. Below is a list of minor-to-moderate issues spotted in the current manuscript.

**We really thank the Reviewer for the positive general comment and for the suggestions provided below. Hereafter, our point-by-point responses to the Reviewer's comments are provided in bold green.**

**Firstly, it is worth noticing that, considering the suggestions of the other two Reviewers and having received a positive answer by the Topic Editor on this, we will prepare a revised version of the manuscript divided in two parts.**

Spotted issues

1. Page 1 –

^ Line 2: Consider replacing "one of" by "among" or something similar,

since "algorithms" is the subject.

**We propose to modify the sentence as follows:**

**"ensemble algorithms are among the most successful data assimilation approaches".**

^ Line 20: What does "a higher order of convergence" mean here?

2. Page 2 –

^ Line 42: "Montecarlo" → "Monte Carlo".

**We thank the Reviewer for spotting these typos. The manuscript will be corrected with "a higher order of approximation" and changing "Montecarlo" into "Montecarlo".**

⌃ Line 51 – 52: Rephrase the sentence "the second order approximation is more effective the closer the ensemble members are to each other, thus, the larger the ensemble spread the worse will be the approximation error in the mean computation."

**According to this comment and to comments from Reviewer #1 and #3, this sentence will be reformulated as follows:**

**"At the same time, most of the models used in geoscience applications are based on systems of differential equations that cannot be represented by a second order polynomial and in all of these cases the second order sampling methods provide a non-exact estimation of the mean, affected by an error proportional to the second order approximation error of the model. Furthermore, the second order approximation is more effective the closer the ensemble members are to each other (i.e., small ensemble spread), thus, the higher the uncertainty the worse will be the approximation error in the mean computation. Since the state estimation is often affected by a relatively high uncertainty in data assimilation geoscience applications, this approximation error may be not negligible."**

3. Line 58 – 59, Page 3: The "2r + 1 ensemble members" requirement does not appear exact. For the unscented transform, one can use either principal component analysis (PCA) or truncated singular value decomposition (TSVD) to reduce the number of ensemble members, see

https://doi.org/10.1175/2008JAS2681.1

https://doi.org/10.1016/j.physd.2008.12.003

It may be worth discussing the similarities and differences between the ideas used to control the number of ensemble members in the aforementioned works and the current manuscript.

**In the mentioned works, the authors reduce the number of ensemble members, which is still 2r+1, by reducing the subspace dimension r (by a PCA or a TSVD). We thank the Reviewer for suggesting the 2 relevant citations which will be added to the introduction.**

4. Line 109 – 117, Page 5: The discussion on the extension of GHOSH to more generic distributions makes sense. A missing part, however, is that the authors did not explain why they confine themselves to Gaussian distribution in the current work, and what could be the challenges for the generalization of GHOSH to more generic distributions

**We thank the Reviewer for this suggestion. We propose to add the following sentence in the Discussion:**

**"Also the statistical moments μ in equation (1) are key hyper-parameters that describe the error pdf moments for orders higher than 2. In our experiments we used the moments of a Gaussian distribution, but it is worth noticing that the GHOSH algorithm does not enforce this choice and other pdfs could be used. However, Kalman filter's**

analysis equations somehow prescribe a Gaussian approximation, thus the GHOSH filter keeps a link to Gaussianity, even if the GHOSH sampling does not. Interestingly other filters, like Hodyss (2011), try to overcome this limitation and could represent good candidates to study the effects of a non-Gaussian GHOSH sampling."

5. Eq. 38, Page 13: The notation w.r.t the Qi component is somewhat confusing. I guess it should be $(Q_p)^{-1}$, but it looks like $Q_{p-1}i$.

Equation 38 will be improved by adding parentheses as suggested.

6. In the experiments w.r.t the Lorenz96 model, localization does seem used. What is the reason behind this setting?

Localization is a strategy to reduce the degrees of freedom in order to successfully apply an ensemble filter when the number of ensemble members is much smaller than the dimension of the state vector. We did not use localization in the Lorenz96 model, since the number of variables was already comparable with the tested ensemble size (at least in the case of ensemble sizes 31 and 63, while the smallest ensemble size settings were specifically aimed to study the behaviour of the filters in undersampling conditions). We propose to add the following sentence to the Filter setup section:

"No localization has been applied, since the number of variables N is relatively small and comparable with some of the ensemble size settings."

7. Line 434 – 435, Page 19: Why "it implies that the PCA measures the Pearson correlation"?

The statement was misleading. We meant that, after a normalisation (i.e., dividing by the standard deviation), the ensemble covariance matrix and the correlation matrix are the same. And, since $\Lambda$ is a diagonal scaling matrix containing the variance of the ensemble, the left hand side of equation (7) becomes equivalent to a renormalization of the basis of the ensemble, i.e.,

$$L^T \Lambda^{-1} L = (\Lambda^{-1/2} L)^T (\Lambda^{-1/2} L),$$

where $(\Lambda^{-1/2} L)$ corresponds to the basis L divided by the standard deviation.

Thus, the PCA is applied to the correlation matrix, and the principal components represent the most relevant correlations.

We propose to correct the statement as follows: "it implies that the PCA is computed on the Pearson correlation matrix".

8. Line 490, Page 21: If I've understood correctly, the "best" label corresponds to the configuration that leads to the best DA performance. If so, then in Figure 4 one should use one block to represent it, and I don't see the point to use a single row for the representation.

**Often, in realistic applications, the number of ensemble members and the observation frequency are not flexible parameters (the former is usually limited by the computational resources and the latter by data availability). On the other hand, the forgetting factor can be tuned to improve assimilation results. Thus, we believe that it could be informative for some readers to see the comparison between the two filters tuned with their best forgetting factor in a range of ensemble sizes and observation frequencies. Considering also a comment from Reviewer 3, we propose to change line 490 to better clarify that "best" means "best RMSE", i.e.,**

**"The first line in each colour-map, labelled "best", represents the best result, in terms of lowest RMSE, obtained among the set of tested forgetting factors."**

9. Line 661, Page 33: "an higher" → "a higher".

**Thank for spotting the typo, that will be corrected.**

**References**

**Hodyss, D., 2011: Ensemble State Estimation for Nonlinear Systems Using Polynomial Expansions in the Innovation. Mon. Wea. Rev., 139, 3571–3588, https://doi.org/10.1175/2011MWR3558.1.**